# Where is the Truth?
# The Risk of Getting Confounded in a Continual World

**Florian Peter Busch** [* 1 2]   **Roshni Ramanna Kamath** [* 1 2]   **Rupert Mitchell** [1 2]
**Wolfgang Stammer** [1 2]   **Kristian Kersting** [1 2 3 4]   **Martin Mundt** [5]

## Abstract

A dataset is confounded if it is most easily solved via a spurious correlation which fails to generalize to new data. In this work, we show that, in a continual learning setting where confounders may vary in time across tasks, the challenge of mitigating the effect of confounders far exceeds the standard forgetting problem normally considered. In particular, we provide a formal description of such *continual confounders* and identify that, in general, spurious correlations are easily ignored when training for all tasks jointly, but it is harder to avoid confounding when they are considered sequentially. These descriptions serve as a basis for constructing a novel CLEVR-based *continually confounded* dataset, which we term the ConCon dataset. Our evaluations demonstrate that standard continual learning methods fail to ignore the dataset's confounders. Overall, our work highlights the challenges of confounding factors, particularly in continual learning settings, and demonstrates the need for developing continual learning methods to robustly tackle these.

## 1. Introduction

Confounding factors represent a dangerous issue for AI systems, often steering models toward shortcuts instead of true understanding (Lapuschkin et al., 2019; Schramowski et al., 2020; Steinmann et al., 2024). Data is considered confounded when models learn spurious correlations that work

---
[*]Equal contribution   [1]Computer Science Department, TU Darmstadt, Germany.   [2]Hessian Center for AI (hessian.AI), Germany.   [3]German Research Center for AI (DFKI), Germany   [4]TU Darmstadt, Centre for Cognitive Science, Germany.   [5]University of Bremen, Department of Mathematics and Computer Science, Germany. Correspondence to: Roshni Ramanna Kamath <roshni.kamath@tu-darmstadt.de>, Martin Mundt <mundtm@uni-bremen.de>.

*Proceedings of the 42nd International Conference on Machine Learning*, Vancouver, Canada. PMLR 267, 2025. Copyright 2025 by the author(s).

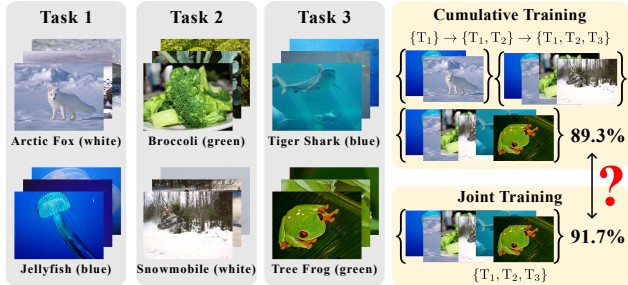

*Figure 1.* **Challenges of continual confounding.** Each task is curated with spurious color correlations potentially affecting a model in learning underlying discriminative features. The different color-based confounding features across sequential tasks demonstrate the impact on the classification problem with joint and cumulative training, with a 2.4% gap in performance.

well during training but fail to generalize to test cases (Ye et al., 2024). This can lead to undesired behavior at deployment both in terms of trustworthiness (*e.g.*, does the model make decisions for the "right" reasons (Ross et al., 2017)) and overall generalization failure (*e.g.*, does the model still make correct predictions (Geirhos et al., 2020)). However, these settings so far have mainly been investigated in conventional, static learning settings (Wu et al., 2023; Sagawa et al., 2020), rather than more realistic, continual learning (CL) scenarios (Chen et al., 2018) where models process data sequentially, creating a fundamental constraint that they cannot access all data simultaneously (McCloskey & Cohen, 1989; Parisi et al., 2019). While prior research indicates that task ordering has minimal impact in such sequential training (De Lange et al., 2022), the distribution of confounders across tasks can impact model performance negatively.

For example, consider an image classification problem on natural images, i.e., ImageNet (Deng et al., 2009). We conducted a controlled experiment using the ResNet-18 model (He et al., 2016) on selected classes (*cf.* Fig. 1). We start by training the model on the first two classes, specifically *arctic fox* and *jellyfish*. We then introduce two additional classes, namely *broccoli* and *snowmobile*, followed by *tiger shark* and a *tree frog*. Note that each class is highly

correlated with specific colors: *white*, *blue*, or *green*. We observe that such a cumulative training setup achieves an average accuracy of 89.3% (across 5 random seeds). Interestingly, when trained on all the data simultaneously *i.e.*, joint training, the model yields an accuracy of 91.7%. This performance gap of 2.4% is substantial, despite utilizing the same model and hyperparameters, and the last training iteration of the cumulative setup using the exact same data as joint training.[1] As images in the first task contain predominantly white or blue backgrounds, we posit that the temporal accessibility constraints give rise to spurious correlations that a model could use for its predictions. These are hard to remedy, as CL limits any access to past observations. Such correlations are unlikely to be similarly challenging when presented in a joint large dataset from the start — order becomes imperative. We posit that the challenges posed by confounding, especially in continual learning settings where data is not accumulated completely, exceed the standard problems usually considered in continual learning and confounding studies separately.

To this end, we systematically investigate model behavior in different confounding settings and show that machine learning (ML) models particularly struggle with *continual confounding*. We observe that this is so *even if training on the joint dataset successfully prevents confounding*; a phenomenon we will later refer to as *insidious continual confounding*. Thus, motivated by the observed disparities in the previous ImageNet-based evaluation and the need for a controlled, systematic analysis, we first introduce a formal description that characterizes continual confounding and highlights its unique challenges. We next introduce ConCon, a controlled confounded visual dataset for continual learning based on CLEVR (Johnson et al., 2017). In ConCon, there exists one ground truth rule which can be used to successfully classify all images. Additionally, each task has one task-specific confounder that can be used to solve the problem for that specific task. For any other task, the previous confounders are not informative. Ideally, an ML model should be able to extract the true rule underlying all tasks instead of focusing on the task-specific confounders. We evaluate the performance of ML and CL methods on this problem and highlight that indeed these focus on the confounders. Moreover, they are unable to extract the ground truth rule, thereby failing to make correct predictions on unconfounded data. Expanding on this general concept of *continual confounding*, we finally empirically corroborate the greater challenge of insidious continual confounding, where a model trained jointly on all tasks can succeed at learning the ground truth rule, while at the same time failing to do so when all data is accumulated sequentially.

---

[1]We include another experiment on a larger number of ImageNet classes in Appendix E, where the gap between joint and cumulative training is even larger.

In summary, our contributions are: i) We design a logical framework that allows us to systematically study *(insidious) continual confounding*, ii) We use the logical framework to construct the ConCon dataset that manifests continual confounding and iii) We empirically evaluate various ML and CL methods and show that they get (insidiously) confounded when trained on ConCon. We publish the ConCon dataset, `https://zenodo.org/doi/10.5281/zenodo.10630481`, and source code for generating the dataset as well as experimental results of our work, `https://github.com/ml-research/concon`.

## 2. Related Work

**Confounding.** The need for trustworthy AI models has led to the curation of several confounded datasets (Steinmann et al., 2024) that allow for a tailored analysis of model behavior. Many of these datasets lie in the visual domain, and due to both the ease of data curation itself, but potentially also to more controlled data settings, existing datasets are typically of a synthetic nature. Important mentions are Toy-Color (Ross et al., 2017), ColorMNIST (Rieger et al., 2020; Kim et al., 2019), Decoy-MNIST (Ross et al., 2017) and CLEVR-Hans3 & 7 (Stammer et al., 2021). However, confounded datasets based on natural images and confounders have also been introduced, *e.g.*, the confounded hyperspectral plant dataset of Schramowski et al. (2020), the ISIC Skin Cancer dataset (Codella et al., 2019; Tschandl et al., 2018), Covid-19 (DeGrave et al., 2021) and CUB5$_{box}$ (Bontempelli et al., 2023). An issue that can occur when analyzing AI models based on natural data versus a controlled synthetic environment is that multiple spurious correlations can occur, such that revising a model to ignore one correlation does not guarantee that it is free from the influence of further, possibly unknown ones (Friedrich et al., 2023). This makes synthetic datasets attractive alternatives, particularly for initial investigations of models and potential mitigation strategies. Despite their great utility in analyzing offline learning settings, none of the confounded datasets mentioned above allow for investigating AI models in terms of confounding factors in the presence of change over time.

**Continual Learning.** The challenge for enabling ML models to adapt to new concepts while retaining previously acquired knowledge has led to the emergence of continual learning (Chen et al., 2018; Mundt et al., 2023b). This has resulted in the creation of a plethora of many sequential dataset benchmarks for studying continual learning (Koh et al., 2021; Peng et al., 2019). Despite several nuanced variations in set-up and evaluation focus (Mundt et al., 2022), the shared objective of all these datasets is to empower the conception of continual techniques that alleviate the catastrophic interference (McCloskey & Cohen, 1989; Ratcliff, 1990). Here, it is assumed that it is strictly beneficial to

retain previous experience as much as possible - to avoid forgetting - without the need to store all prior observed data (Hayes et al., 2021; Kudithipudi et al., 2022). Respective benchmarks typically convert existing datasets into disjoint incremental variants (*e.g.*, permuted MNIST (Matan et al., 1990; Goodfellow et al., 2015), or class incremental CIFAR-10 (Krizhevsky & Hinton, 2009) and ImageNet (Deng et al., 2009)), successively learn on individual (language) datasets (Biesialska et al., 2020), or simulate controlled distribution changes (Hess et al., 2021). Recently, neuro-symbolic continual CLEVR (Marconato et al., 2023) followed in this spirit of devising a disjoint sequence of tasks, introducing the first dataset with clear underlying logical rules. However, it follows prior art in focusing on incrementing, *e.g.*, object type or color to highlight the role neuro-symbolic methods may play in knowledge accumulation. Existing continual learning datasets, while useful for investigating sequential adaptation, do not support the systematic analysis of varying confounders across tasks. With ConCon, we introduce the first CL dataset where confounders differ across tasks.

## 3. Continual Confounding

Recall the introductory example with changing confounders over time. While each point in time (task)[2] is clearly confounded when considered separately, it will turn out that subtleties in the specification used when generating these confounded datasets can have drastic consequences for the extent to which a subset of tasks remains confounded when considered jointly. Two possible choices for such specifications are shown in Figure 2. In line with confounders often encountered in reality, *e.g.*, specific watermarks, the first specification, labeled `disjoint` in the upper half of Figure 2, requires that a confounder is only observed in its respective task. This makes finding a simultaneous solution to all tasks easier, since learned behavior specific to the presence of the confounder from one task need not be unlearned in order to solve some other task. We will later demonstrate with this setting that the avoidance of catastrophic forgetting across differently confounded tasks is insufficient to ensure acceptable generalization to unconfounded data.

The `strict` specification (lower half of Figure 2), on the other hand, has no such requirements regarding the presence or labeling of confounders from one task in the examples of another. This results in a much greater disparity between behavior when training separately versus jointly on the tasks, since images satisfying the confounding rule of one task may appear in the negative set of some other task. In fact, our later evaluations in this setting may challenge traditional CL wisdom with respect to memory rehearsal as a solution.

### 3.1. Formal Description of Continual Confounding

Let us now provide a formal description of the aforementioned settings to support systematic data generation. Let $g$ refer to a ground truth predicate, which maps images to a boolean classification. Let $c_t$ refer to the $t$-th confounding predicate. For a task $t$, we have a set of positive examples $\mathcal{P}_t$ and a set of negative examples $\mathcal{N}_t$. $\mathcal{P}_t$ and $\mathcal{N}_t$ are drawn randomly from examples satisfying the defining predicates $p_t$ and $n_t$ respectively. When we say the task is confounded with confounder $c_t$, we mean that the task can be solved both with the ground rule $g$ and with $c_t$. From this we may derive the following constraints on $p_t$ and $n_t$: $p_t \leq g \wedge c_t$, $n_t \leq \neg(g \vee c_t)$, where $\wedge$ and $\vee$ are logical conjunction and disjunction respectively, and we define the partial ordering $x \leq y$ if and only if $x \implies y$.[3] Intuitively, these conditions state that all examples in $\mathcal{P}_t$ satisfy both $g$ and $c_t$, and all examples in $\mathcal{N}_t$ satisfy neither $g$ nor $c_t$. The natural choice

$$p_t := g \wedge c_t, \quad \neg n_t := g \vee c_t \quad (1)$$

is the most permissive regarding which examples are valid dataset members, and is therefore least permissive regarding the simultaneous solution of all tasks. We accordingly call this dataset choice `strict`. The reader will note that, in this case, the confounder $c_t$ of task $t$ may in general occur in both the positive and negative set of other tasks $t'$. We define our second dataset, `disjoint`, such that confounders $c_t$ are observed only in their own task, *i.e.*, in neither the positive $\mathcal{P}_{t'}$ or negative $\mathcal{N}_{t'}$ sets of any other task $t'$. We do this both because such datasets are encountered in practice and because it results in interesting consequences for the possible joint solutions of all tasks, as will be discussed shortly. The constraints defining the negative set of the `disjoint` dataset are

$$p_t := g \wedge c_t \wedge \bigwedge_{i=1, i \neq t}^{T} \neg c_i, \quad \neg n_t := g \vee \bigvee_{i=1}^{T} c_i \quad (2)$$

where we have used the summation-style notation $\bigvee_{i=1}^{T}$ to indicate disjunction over all $i$ for $n_t$, and $\bigwedge_{i=1, i \neq t}^{T}$ to indicate conjunction over all negations of confounders from tasks not equal to $t$ for $p_t$.

Let us now consider the implications for the predicate $r$ learned by our network for the tasks both jointly and separately. Assuming that the learned predicate $r$ achieves perfect accuracy on the sets $\mathcal{P}_t$ and $\mathcal{N}_t$ from some task $t$, we obtain the following constraints on $r$:

$$p_t \leq g \wedge c_t \leq r \leq g \vee c_t \leq \neg n_t. \quad (3)$$

---

[2]We consider $t$ to be a time step, but to be consistent with the norm of CL literature, we refer to $t$ as a task, even if the objective remains the same, but the underlying data distribution changes.

[3]A boolean algebra interpreted this way forms a distributive lattice, *i.e.*, a partially ordered set with least upper bound $a \vee b$, greatest lower bound $a \wedge b$, and in which $\vee$ and $\wedge$ distribute over each other.

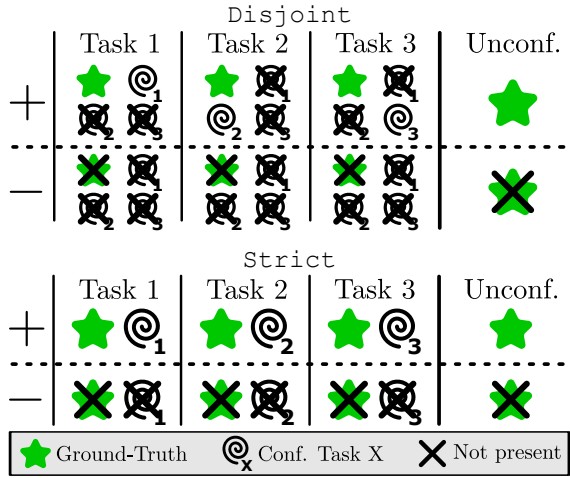

| | Ground-Truth | | Conf. Task X | | Not present |

*Figure 2.* **Abstract Representation of the ConCon Dataset.** The unconfounded dataset (rightmost column) is generated using only the ground truth rule (★). For all three tasks, this rule is still present in positive and absent in negative images. *All* positive images contain the confounded object (◎). For the `strict` dataset, only the respective confounders are avoided in the negative samples. In the `disjoint` dataset, the confounders of *any* task may not reappear in either positive or negative images.

To solve all tasks jointly, $r$ must satisfy $T$ such constraints simultaneously. Unless all confounders $c_t$ are the same, this combined constraint will differ from the individual constraints, a situation we will refer to as *continual confounding*. This can be expressed as a least upper bound (resp. greatest lower bound) for the left (resp. right) hand side of the constraints. In the `strict` case this gives

$$\bigvee_{t=1}^{T} (g \wedge c_t) \leq r \leq \bigwedge_{t=1}^{T} (g \vee c_t), \tag{4}$$

simplifying to

$$g \wedge \bigvee_{t=1}^{T} c_t \leq r \leq g \vee \bigwedge_{t=1}^{T} c_t \tag{5}$$

by distributivity. Intuitively, this means that if $g$ is true and at least one $c_t$ is true, then the learned predicate $r$ must evaluate to true. And, dually, that if $r$ is true then either $g$ must be true, or all confounders $c_t$ must be true.

Repeating the above process for `disjoint`, we obtain

$$g \wedge \bigvee_{t=1}^{T} c_t \leq r \leq g \vee \underline{\bigvee_{t=1}^{T}} c_t \tag{6}$$

where $\underline{\bigvee}_{t=1}^{T} c_t$ is the exclusive disjunction over tasks $t$, *i.e.*, exactly one of the $c_t$ must be true. It can be seen that both sides of the constraints have been weakened, from

disjunction to exclusive disjunction on the left, and from conjunction to disjunction on the right. These are both weakenings of the constraints on $r$ since $\underline{\bigvee}_{t=1}^{T} c_t \leq \bigvee_{t=1}^{T} c_t$ and $\bigwedge_{t=1}^{T} c_t \leq \bigvee_{t=1}^{T} c_t$. Intuitively, we now require that if $g$ is true and exactly one $c_t$ is true, then the predicate $r$ must evaluate to true. And if $r$ is true then either $g$ must be true, or at least one confounder $c_t$ must be true.

**Logical Implications.** By studying these conditions, we may make several interesting observations about the possible behavior of our learner. Firstly, as a result of the weakened right hand side, $r = \bigvee_{t=1}^{T} c_t$ is a valid solution to the `disjoint` case, (as can be seen by substitution into Equation 6). This corresponds to the fact that the rule "example is positive if it satisfies the confounder rule from any task we have seen so far" is a valid solution at all points in training. In combination with the weakened left hand side, it is additionally true that the learned rule may have arbitrary behavior in the simultaneous presence of multiple confounders. This corresponds to the fact that, in the `disjoint` setting, multiple confounders may never occur together in the same example during training. Importantly, none of these solutions are likely to be valid for the unconfounded test (which used only $g$ for data generation) since none of them even mention $g$.

As we will later see empirically, the `strict` case is less vulnerable to this variety of pathology. Intuitively, this is because any predicate $r$ jointly satisfying all task constraints must either equal $g$ or contain within itself a statement of $g$, and thereby have less prior probability than $g$. In particular, none of the above trivial solutions to the `disjoint` case generalize to the `strict` one. It is also possible in the `strict` case to force a tighter bound on $r$ by restricting the choices of $c_t$. If the confounders are exhaustive, *i.e.* $\bigvee_{t=1}^{T} c_t = \top$, then there can be no example which satisfies $g$ but not $r$, since $g = g \wedge \top \leq r$, where $\top$ denotes logical truth. Dually, if it is impossible to simultaneously satisfy all confounders, *i.e.* $\bigwedge_{t=1}^{T} c_t = \bot$, then there can be no example which satisfies $r$ but not $g$, since $r \leq g \vee \bot = g$, where $\bot$ denotes logical falsity. When both of these conditions are satisfied we have $g \leq r \leq g$, and the ground truth predicate $g$ is the unique joint solution to all tasks.

**Insidious Continual Confounding.** An interesting property of the `strict` dataset is that while each task is by definition confounded when considered in isolation, the joint dataset is likely to have the ground truth $g$ as its *a priori most probable solution*. As we later see empirically, this means that, if presented with all tasks simultaneously, our learner is likely to correctly infer the ground truth rule.

In principle, a continual learner could remember sufficient information from earlier tasks to reconstruct the ground truth rule $g$ once it has seen all tasks. However, continual learners are generally designed to retain information *only*

*relevant for decision making* from previously observed tasks. If the continual learner has solved early tasks by exploiting confounders, it will likely not have retained information sufficient to derive the ground truth rule. To such a continual learner, the joint task will simply seem inconsistent and impossible: it perfectly remembers its decision rules for the perfect solution of earlier tasks, and yet they are incompatible. This occurs despite the fact that, if it had initially been presented with the full joint dataset containing all tasks, the learner would have been able to derive the ground truth rule. Furthermore, even if there is in principle sufficient information remembered from earlier tasks for the continual learner to reject the confounded decision rules, we will later see that continual training while accumulating all data seen so far (*i.e.*, an infinite memory buffer) may still fail where joint training succeeds. Since this variety of confounding *only* occurs once the continual aspect of the problem is introduced, we refer to it as *insidious continual confounding*, a specific kind of continual confounding.

### 3.2. The ConCon Dataset

Based on the previous descriptions, we now introduce the ConCon dataset. In general, every data point is an image containing multiple objects with several attributes, building on the framework of CLEVR (Johnson et al., 2017). New data is simulated using the Blender software, and the ground truth rules and confounders are specified in JSON files. For our `disjoint` and `strict` ConCon dataset variants, we have defined 3 tasks based on a common ground truth rule, and each task has been assigned a unique confounder. The presence of the ground truth rule determines the binary class assignment of each image. An image can be either positive or negative (*i.e.*, ground truth rule is or is not satisfied). In this continual setting, different tasks vary by the presence or absence of confounders, whereas the class that should be predicted remains the same (see Figure 2). Specifically, we have chosen each image to contain 4 objects, with a choice among 2 sizes, 2 materials, 3 shapes, and 8 colors. In addition, a universal unconfounded dataset contains images that are based only on the ground truth rule. Further details, including a dataset sheet, are provided in the appendix.

For our `disjoint` and `strict` ConCon variants, the exact ground truth rule was chosen as: *sphere and small cube*, see Figure 3. The *task-specific confounding rules* are *blue*, *metal*, and *large*, respectively. In the `strict` setting, all positive images of a task satisfy the ground truth rule and contain the task-specific confounder. For example, for the first task, all positive images contain both a *sphere and small cube* and a *blue* object. Conversely, the negative images of a task neither satisfy the ground truth rule nor contain the task-specific confounder.

In the `disjoint` case, the positive images of a task not

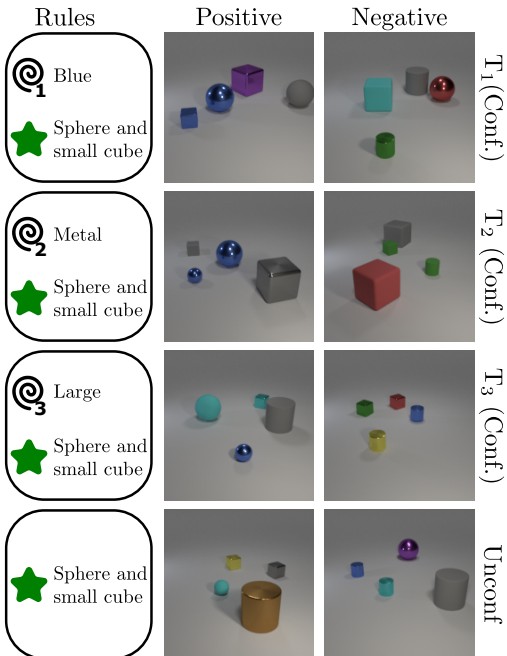

*Figure 3.* **ConCon samples.** Shown are positive and negative examples from our `strict` dataset. Throughout all tasks ($T_1$, $T_2$, $T_3$), a *sphere and a small cube* (ground truth rule (★)) must be present in a positive image but can not appear together in a negative image. Alternatively, each of the three tasks may be solved by exploiting a confounder (presence of a *blue*, *metal*, or *large* object, respectively (◉)). The contents of the unconfounded images depend only on the ground truth rule.

only satisfy the ground truth and contain the task-specific confounder, but they further do not contain confounders specific to other tasks. Hence, for the first task, all positive images contain a *sphere and small cube* and *blue* object and no *metal* or *large* object, all positive images in the second task contain a *sphere and small cube* and *metal* object and no *blue* or *large* object, and all positive images in the third task contain a *sphere and small cube* and *large* object and no *blue* or *metal* object. The negative images contain neither *sphere and small cube* nor *blue* nor *metal* nor *large*, across all the tasks. All other object properties are randomly sampled.

An important consideration in ConCon is the distribution shifts caused by the different confounding features and how they change across tasks. Specifically, features that act as confounders in one task can also appear as random features in other tasks. The distribution shift between the confounded tasks and the unconfounded dataset is a result of such features no longer acting as confounders, *i.e.*, they might appear on both positive and negative images but are not informative with respect to the class label. Our synthetic data generation used for ConCon ensures minimal distribution shift between confounded and unconfounded datasets while still exhibiting confounding in one but not the other. Thus, the resulting

minimal train-test mismatch is intentional and allows for investigating whether a high model accuracy is caused by the model making predictions based on the confounding features or the ground truth features.

Following the specifications outlined above, we have systematically generated a comprehensive dataset comprising of 3000 train, 750 validation, and 750 test samples for both positive and negative sets of each task resulting in 9000 images per task. The additional unconfounded dataset is generated with the same size as these splits. Overall, a total of 63000 images have thus been generated for the ConCon dataset to enable thorough investigation. Precise implementation details are provided in the appendix.

In summary, ConCon is the first dataset to support the systematic investigation of the role of confounders in CL. The synthetic domains prove especially valuable in isolating and analyzing how different training mechanisms interact with data distribution characteristics. More importantly, ConCon allows us to conceive even more challenging settings, where the reader may perhaps be surprised to find that the introduction of the continual element alone is sufficient to produce confounding. And furthermore, this confounding, which was not otherwise present, even persists when sequentially accumulating the dataset with infinite memory.

## 4. Experimental ConCon Evaluation

We now investigate whether continual learning methods overcome the confounding pitfalls of ConCon. Specifically, we aim to answer the following two questions:

**Q1**. Does preventing catastrophic forgetting help with continual confounding?

**Q2**. Do continual learning methods suffer from insidious continual confounding?

### 4.1. Setup and Evaluation

**Models.** For each experiment, we use a ResNet-18 model (He et al., 2016) (henceforth referred to as neural network or NN) and the neuro-symbolic concept learner (Stammer et al., 2021) (henceforth simply called NeSy) that is based on the Slot Attention module (Locatello et al., 2020) and Set Transformer (Lee et al., 2019). Where the ResNet-18 model bases its decisions on learned visual features, the NeSy model first extracts object property representations before making a final prediction. We describe training details and hyperparameters for both models in Appendix B.

**Sanity Check.** To investigate whether a model is able to generalize to the unconfounded dataset after training on the confounded dataset, it must be established that the models can succeed in learning the ground truth rule when no confounders are present. To this end, we train both models only

on the unconfounded dataset where the only valid solution is the ground truth rule itself. For the NN, we report an accuracy of $96.96 \pm 0.44$ and NeSy achieves $80.33 \pm 0.91$ accuracy. This confirms that the models can successfully learn the ground truth rule in principle.

**Comparison of Methods.** We investigate eleven methods on ConCon. First, we train both models on the entire confounded dataset, where we would like to see the interplay of the confounding rules across all tasks in *joint training*. As a complement, we evaluate the model performance when trained on the 3 tasks one after the other in a *naive sequential* manner. To combat catastrophic forgetting expected in the latter, we use memory-based CL methods such as ER (Chaudhry et al., 2019), DER (Buzzega et al., 2020), and BGS (Lee et al., 2024), the regularization-based method EWC (Kirkpatrick et al., 2017), and FROMP (Pan et al., 2020) which is a mix of both. We also evaluate the performance of GDUMB (Prabhu et al., 2020) on ConCon that is competitive against numerous CL methods and of PNNs (Rusu et al., 2022), where the model is expanded with each new task. We also evaluated a generative replay method (van de Ven et al., 2020), results of which can be found in Appendix D. For all the methods requiring a memory buffer, we fix the size to 100 and populate it according to what each method stipulates. Finally, we investigate a *cumulative* and a *shuffled* baseline. In the cumulative case, we append the entire dataset from previous task(s) to the current one and continue training the model on the dataset accumulated so far. This is typically treated as an upper-bound to CL performance in the spirit of ER with infinite data memory (Hayes et al., 2021), which will be questioned in our later findings.[4] To isolate the consequences of confounding, in the shuffled setup, all the data is shuffled randomly across tasks, *i.e.*, the confounders are no longer aligned with the tasks. This breaks the temporal alignment of the confounders while maintaining the CL constraint of the cumulative setup. We provide more detailed information on the CL methods and the full training details in Appendix B.

**Evaluation Metrics.** We measure the accuracy on the confounded test datasets across all tasks after training the model on each task. To investigate whether the model gets confounded, we further measure the accuracy on the unconfounded test set. We omit the standard deviations, as they are negligible (we include further results on the standard deviations in Appendix C).

### 4.2. CL Methods Get Continually Confounded

We start by investigating the ConCon `disjoint` case, with evaluation results featured in the Table 1. Results on both

---

[4]Since this is the largest possible buffer size, applying buffer-based methods like ER with larger buffer sizes is expected to yield results between what we run and the cumulative setup.

*Table 1.* **ConCon Evaluation on `disjoint`.** Test accuracies on `disjoint` for NN and NeSy. Current task (T) performance is high almost throughout. Naive sequential training and EWC suffer from forgetting, showcasing a huge drop in performance on previous tasks. While other methods manage to keep high accuracies on previous tasks, our results on `disjoint` show that avoiding forgetting does not help with performance on unconfounded data. This holds true for both NN and NeSy.

| | Method | Current Task Acc. | | | Old Task Acc. | | Unconf. |
|---|---|---|---|---|---|---|---|
| | | $T_1$ | $T_2$ | $T_3$ | $T_1@T_3$ | $T_2@T_3$ | |
| **NN** | Naive Sequential | 100.0 | 92.2 | 100.0 | 50.0 | 63.6 | 48.1 |
| | EWC (Kirkpatrick et al., 2017) | 100.0 | 99.9 | 100.0 | 50.1 | 60.6 | 47.7 |
| | FROMP (Pan et al., 2020) | 100.0 | 99.8 | 99.4 | 99.2 | 97.9 | 50.2 |
| | ER (Chaudhry et al., 2019) | 100.0 | 99.6 | 100.0 | 100.0 | 99.6 | 50.1 |
| | DER (Buzzega et al., 2020) | 100.0 | 99.9 | 100.0 | 99.2 | 98.1 | 50.2 |
| | BGS (Lee et al., 2024) | 100.0 | 99.9 | 100.0 | 95.1 | 96.2 | 50.4 |
| | GDUMB (Prabhu et al., 2020) | 64.1 | 62.0 | 69.5 | 63.6 | 61.2 | 48.8 |
| | PNNs (Rusu et al., 2022) | 100.0 | 99.8 | 100.0 | 100.0 | 99.8 | 47.9 |
| | **Shuffled** | **100.0** | **99.9** | **100.0** | **100.0** | **99.8** | **50.0** |
| | **Cumulative** | **100.0** | **100.0** | **100.0** | **100.0** | **99.9** | **49.9** |
| | **Joint** | **100.0** | **99.9** | **100.0** | **N/A** | **N/A** | **50.0** |
| **NeSy** | Naive Sequential | 99.5 | 99.9 | 100.0 | 50.0 | 58.1 | 50.0 |
| | EWC (Kirkpatrick et al., 2017) | 99.6 | 100.0 | 99.9 | 50.2 | 67.2 | 50.2 |
| | FROMP (Pan et al., 2020) | 99.6 | 98.9 | 98.4 | 97.9 | 97.3 | 50.5 |
| | ER (Chaudhry et al., 2019) | 99.5 | 99.8 | 99.9 | 99.2 | 99.3 | 50.7 |
| | DER (Buzzega et al., 2020) | 99.6 | 99.7 | 99.9 | 98.6 | 96.9 | 50.9 |
| | BGS (Lee et al., 2024) | 99.6 | 99.9 | 99.9 | 94.4 | 95.1 | 50.9 |
| | GDUMB (Prabhu et al., 2020) | 99.4 | 97.8 | 98.5 | 98.0 | 98.2 | 50.3 |
| | PNNs (Rusu et al., 2022) | 99.5 | 100.0 | 100.0 | 99.6 | 100.0 | 49.6 |
| | **Shuffled** | **99.8** | **99.7** | **99.9** | **99.7** | **99.9** | **50.8** |
| | **Cumulative** | **99.8** | **99.9** | **99.9** | **99.8** | **99.9** | **50.8** |
| | **Joint** | **99.7** | **99.9** | **99.9** | **N/A** | **N/A** | **50.7** |

(The left margin label `disjoint` spans the full table.)

NN and NeSy are reported and grouped into three categories per method: current task accuracy, old task accuracy, and accuracy on the unconfounded dataset.

For both NN and NeSy, we can see that most models successfully classify the current tasks. However, the performance of the "Naive Sequential" approach on old tasks illustrates the necessity of continual learning methods to avoid forgetting. While EWC does not appear to significantly alleviate forgetting, we can see the memory-based methods ER, DER, BGS, and FROMP (benefiting mainly from the stored past examples), as well as PNNs (benefiting from additional classification heads) doing so successfully, thereby achieving close to 100% accuracy on *all confounded* tasks after having learned them once. GDUMB trains the model using only the buffer, resulting in bad performance for NN but is otherwise in line with the other memory-based approaches. Nevertheless, all methods on both NN and NeSy fail on the unconfounded dataset, even for the "Cumulative" setup, where all previous instances are kept when training the model sequentially. The models get continually confounded. This also remains true in both cases where a model is trained on data from all tasks at the same time, no matter if it is trained continually (shuffled) or on the full dataset at the same time (joint). Since even the joint training setup does not learn the ground truth rule, it is unsurprising that the continual learning methods fail to do so as well.

Our results show that both models prefer learning a combination of the confounded tasks over the ground truth rule, as they provide reliable shortcuts for solving individual tasks during sequential training. As highlighted by our logical analysis, this can be explained by our models learning the disjunction of all confounding features instead of the ground truth rule, despite the models' general ability to learn the ground truth rule, as previously shown in our sanity check.

Ultimately, **does preventing catastrophic forgetting help with continual confounding?** Our evaluation on the `disjoint` dataset showed that, in this setting where learning the disjunction of confounders is preferred over learning the ground truth rule, CL methods can successfully avoid forgetting. However, **preventing forgetting does not help with making correct predictions on unconfounded data**. In particular, the ground truth rule is also not learned when using a maximally large memory buffer (cumulative training that stores all data) or when using the NeSy approach.

*Table 2.* **ConCon Evaluation on `strict`.** Test accuracies on `strict` for NN and NeSy. All continual learning methods fail to achieve high accuracies on previous tasks, especially for NN. Strikingly, even cumulative continual learning fails to find a solution on unconfounded data for `strict` with NN, despite *joint training* on all confounded data succeeding. While *shuffled training* also sees a small drop in accuracy, a big gap to joint training remains. This shows that the majority of the difference between joint and cumulative training is not caused by the CL setting, but by the confounding setup, highlighting the manifestation of *insidious continual confounding*.

|  | Method | Current Task Acc. | | | Old Task Acc. | | Unconf. |
|---|---|---|---|---|---|---|---|
|  |  | $T_1$ | $T_2$ | $T_3$ | $T_1@T_3$ | $T_2@T_3$ |  |
| **strict** / NN | Naive Sequential | 100.0 | 100.0 | 100.0 | 49.1 | 48.9 | 50.0 |
|  | EWC (Kirkpatrick et al., 2017) | 100.0 | 99.9 | 100.0 | 49.1 | 48.9 | 50.0 |
|  | FROMP (Pan et al., 2020) | 100.0 | 59.9 | 69.2 | 67.8 | 59.1 | 51.1 |
|  | ER (Chaudhry et al., 2019) | 100.0 | 97.7 | 96.4 | 54.6 | 59.6 | 51.0 |
|  | DER (Buzzega et al., 2020) | 100.0 | 98.7 | 96.5 | 52.8 | 55.5 | 49.1 |
|  | BGS (Lee et al., 2024) | 100.0 | 99.8 | 99.6 | 50.1 | 49.1 | 49.5 |
|  | GDUMB (Prabhu et al., 2020) | 92.5 | 62.7 | 53.6 | 57.5 | 54.2 | 50.4 |
|  | PNNs (Rusu et al., 2022) | 100.0 | 99.8 | 99.8 | 100.0 | 99.8 | 49.2 |
|  | **Shuffled** | **79.9** | **93.4** | **95.3** | **95.2** | **96.7** | **89.5** |
|  | **Cumulative** | **100.0** | **87.0** | **88.8** | **85.9** | **91.0** | **72.6** |
|  | **Joint** | **98.2** | **99.2** | **98.9** | **N/A** | **N/A** | **95.7** |
| **strict** / NeSy | Naive Sequential | 99.7 | 100.0 | 100.0 | 49.1 | 49.2 | 49.8 |
|  | EWC (Kirkpatrick et al., 2017) | 99.9 | 99.9 | 100.0 | 49.1 | 49.4 | 50.1 |
|  | FROMP (Pan et al., 2020) | 99.8 | 83.4 | 74.7 | 85.1 | 76.3 | 56.2 |
|  | ER (Chaudhry et al., 2019) | 99.8 | 99.8 | 98.8 | 68.0 | 83.6 | 61.2 |
|  | DER (Buzzega et al., 2020) | 99.7 | 99.4 | 99.5 | 58.8 | 73.2 | 56.5 |
|  | BGS (Lee et al., 2024) | 99.8 | 99.9 | 99.9 | 51.8 | 56.3 | 51.3 |
|  | GDUMB (Prabhu et al., 2020) | 99.4 | 85.6 | 82.7 | 80.1 | 84.2 | 56.7 |
|  | PNNs (Rusu et al., 2022) | 99.8 | 100.0 | 100.0 | 99.8 | 100.0 | 49.9 |
|  | **Shuffled** | **87.8** | **92.5** | **89.2** | **88.2** | **92.6** | **70.1** |
|  | **Cumulative** | **99.8** | **92.5** | **88.8** | **88.6** | **92.3** | **69.9** |
|  | **Joint** | **88.4** | **92.6** | **89.8** | **N/A** | **N/A** | **70.8** |

## 4.3. CL Methods Can Get Insidiously Confounded

We have seen that CL methods do aid in alleviating forgetting, but do not help prevent confounding when the *disjunction of confounders* is learned over the ground truth rule on the joint dataset. On our `strict` dataset, we will now empirically corroborate that there in fact exists an even more concerning case: the case where models suffer from being confounded primarily through learning continually.

The experimental results on the `strict` dataset are shown in Table 2. First, consider the only approach that outputs different predictions depending on which data the task is coming from: PNNs. Here, the model has different classification heads for each task, enabling it to fully avoid forgetting and perform well on current and past tasks. However, this leads to learning the respective tasks individually, failing to identify the underlying ground truth rule that applies to all tasks. Considering the more standard continual learning setups, it is again unsurprising that previous tasks appear to be forgotten in the naive sequential approach. The model likely learns and focuses on the new, task-specific confounder at every stage. This may be explained by the

incompatibility of confounders (*i.e.*, applying a rule for a task-specific confounding feature on another task results in no better than random prediction). EWC also shows no significant difference to the naive training setup. However, differences between our dataset variants start to show when considering the memory-based approaches.

Whereas most of these were previously able to mitigate forgetting fully across all tasks in the `disjoint` case, we now observe that ER, DER, and BGS perform not much better than naive sequential training in the `strict` scenario. Importantly, both NN and NeSy models additionally fail to make accurate predictions on the unconfounded dataset. FROMP and GDUMB appear to sacrifice current task performance for old task performance, but this does not improve performance on unconfounded data in our experiments compared to, for example, ER and gives strictly worse results than the cumulative (ER with infinite memory) approach. All standard CL methods thus not only fail to perform well on the unconfounded data but also do not get close to successfully avoiding forgetting, with accuracies on old tasks not even exceeding 60% except for FROMP, where

current task performance is much lower. This shows how confounded datasets can be especially challenging for CL methods, even when those successfully avoid forgetting in easier problems. While the cumulative approach, benefiting from its infinite buffer size, performs much better in comparison, contrasting it with joint and shuffled training on all tasks reveals results that may strike the reader as surprising.

For NeSy, cumulative training and joint training perform similarly well, but they both do not reach their earlier baseline performance on the unconfounded dataset ($70.76\%$ vs $80.33\%$). More interestingly, we find the cumulative training with NN to seemingly focus more on the confounded tasks than on the ground truth, resulting in predictions more than $20\%$ worse than in the joint training setup, despite training on the exact same data. The latter in fact empirically highlights how both NN and NeSy perform significantly better than random guessing when trained jointly on the `strict` dataset. Despite each task being confounded, the NN achieves an astonishing $95.68\%$ accuracy on the unconfounded data, compared to the achievable $96.96\%$ when trained on the unconfounded data directly (recall the initial sanity check). While some of the differences between joint and cumulative training may not be solely explained by the presence of confounders, but also by the continual learning setup itself, we observe in our shuffled experiment that training continually on the same data yields substantially better results when the tasks are not aligned with the confounding features. In other words, we now experimentally observe our prior formalized *insidious continual confounding*. That is, **training in a continual setup performs significantly worse than the joint setup**, challenging the conventional wisdom that continual learning revolves around the accumulation of data and knowledge and questioning the role of cumulative training as a performance upper-bound (Hayes et al., 2021; Mundt et al., 2023b).

In conclusion, we can thus **answer our question "do continual learning methods suffer from insidious continual confounding?" affirmatively**. Our evaluation shows that CL in the presence of confounding is significantly more challenging than preventing forgetting. Both our models *prevent forgetting* in the `disjoint` case, where the solution of all confounded tasks could be combined in a relatively simple manner (disjunction). However, this proved *insufficient to learn the ground truth rule* required to perform as well on the unconfounded data as joint training in the `strict` case.

## 5. Conclusion

We have introduced the first continually confounded dataset, ConCon, and have formally described its implications for systematic analysis. ConCon consists of two variants that implement different types of confounding. In our evaluations, we have empirically demonstrated that even perfect prevention of catastrophic forgetting is insufficient to avoid confounding in `disjoint` settings, where even joint training fails to generalize. Most importantly, we successfully confirmed the latter to not be true for our `strict` dataset, where joint training succeeds but continual cumulative training is veritably insufficient due to the manifestation of *insidious continual confounding*. This stands in stark contrast to the typical assumption that knowledge accumulation is the desirable solution in continual learning.

Our work thus provides the means to benchmark and further explore confounding in CL, for example, by investigating causal relationships (Mundt et al., 2023a) and related innovative new perspectives on, *e.g.*, causal rehearsal (Churamani et al., 2023) or bias mitigation of spurious correlations (Lee et al., 2023). We envision that a deeper understanding of continual confounding can lead to overall better model performance in continual settings. Other modalities, such as text or time series, are not currently in the focus of ConCon and are left for future research.

## Impact Statement

Our paper challenges the conventional view of cumulative learning as an upper bound when evaluating continual learning methods, thereby showing that such confounders have a detrimental effect on model performance even in what is conventionally considered a best-case scenario. The issue of confounders in machine learning represents a significant challenge. Making predictions based on such can not only lead to incorrect behavior but may also lead to unfair or discriminatory decisions (Schröder et al., 2023). In our specific case of ConCon, the dataset consists of synthetically generated geometric shapes and thus does not contain any personal or otherwise private information. At present, we do not envision any scenario in which it could be used with malicious intent. By means of ConCon, this paper aims to improve the evaluation of continual learning methods when confounding might be present, inspire new approaches to deal with these issues, and facilitate systematic investigation of such approaches in the future.

## Acknowledgements

This work is supported by the Hessian Ministry of Higher Education, Research, Science and the Arts (HMWK; projects "The Third Wave of AI" and "The Adaptive Mind"). It further benefited from the Hessian research priority programme LOEWE within the project "WhiteBox" and the EU-funded "TANGO" project (EU Horizon 2023, GA No 57100431). The authors thank Laura Boyette for the preliminary results on this research. Furthermore, the authors would like to thank Felix Divo for proofreading our paper and thus improving the clarity of the final manuscript.

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

# Appendix

This appendix complements the main body with:

**A:** Additional dataset details and its rules for generating positive and negative samples.

**B:** Training information for the NN (ResNet-18) and NeSy (Neuro-Symbolic concept learner) models.

**C:** Training curves to further corroborate that mitigation of forgetting is insufficient to tackle *insidious continual confounding*, together with individual test accuracies on the positive and negative images for deeper empirical investigation of the fallacies.

**D**: Experimental results on the ConCon dataset using a generative model.

**E**: An experiment comparing joint and cumulative training on a dataset comprised of several ImageNet classes.

**F**: A statement on limitations of our paper and the introduced ConCon dataset.

**G**: A datasheet for datasets (Gebru et al., 2021) to document *e.g.*, composition, collection process, and other aspects.

## A. ConCon and the Code Framework

Along with our paper, we also publish the specific ConCon dataset[5], as well as the data generation platform and the training code[6] to facilitate further research on continual confounding. We include a croissant (Akhtar et al., 2024) metadata record in our GitHub repository.[7] ConCon uses the BSD 2-Clause "Simplified" License, and our code for generating the data and running the experiments is released under the MIT license.

### A.1. Dataset Details

Recall that ConCon is built on top of CLEVR (Johnson et al., 2017) and thus consists of images representing geometric objects with different properties. Each image is an RGB image of size $224 \times 224$ depicting four objects. In our dataset, an object can be described by four properties, namely **shape** (*cube*, *sphere*, *cylinder*), **material** (*metal* or *rubber*), **size** (*small* or *large*), and **color** (*gray*, *red*, *blue*, *green*, *brown*, *purple*, *cyan*, *yellow*). Data is generated with respect to the ground truth rule and confounders as described in the main paper. Relationships between objects are not used in ConCon. We show several samples in Figure 4.

As we describe in detail in our main paper, the positive instances in the dataset fulfill the ground truth rule ($g$) and the negative instances do not ($\neg g$). In addition, it is possible that a confounding feature must apply ($c_t$) or must not apply ($\neg c_t$), depending on the task (denoted by subscript $t$) and the dataset variant (`disjoint` or `strict`). The ground truth $g$ is defined as *sphere and small cube*. The confounders, in the order of tasks, are $c_1$: *blue*, $c_2$: *metal*, and $c_3$: *large*. We summarize the rules that should hold for respective tasks in Tables 3 (`disjoint`) and 4 (`strict`).

*Table 3.* **Rules `disjoint`:** a confounding feature may only appear in the positive images of its respective task. A common rule is thus applied for all negative images across all tasks.

|        | Positive | Negative |
|--------|----------|----------|
| Task 1 | $g \wedge c_1 \wedge \neg c_2 \wedge \neg c_3$ | $\neg g \wedge \neg c_1 \wedge \neg c_2 \wedge \neg c_3$ |
| Task 2 | $g \wedge \neg c_1 \wedge c_2 \wedge \neg c_3$ | $\neg g \wedge \neg c_1 \wedge \neg c_2 \wedge \neg c_3$ |
| Task 3 | $g \wedge \neg c_1 \wedge \neg c_2 \wedge c_3$ | $\neg g \wedge \neg c_1 \wedge \neg c_2 \wedge \neg c_3$ |

### A.2. Code Framework

It is possible to generate further instances or make changes to the data generation process by using our code repository. The logic in the form of ground truth and confounders, which determines the data generation, can be set in separate files.

---

[5]https://zenodo.org/doi/10.5281/zenodo.10630481
[6]https://github.com/ml-research/concon
[7]https://github.com/ml-research/concon/blob/main/croissant_concon.json

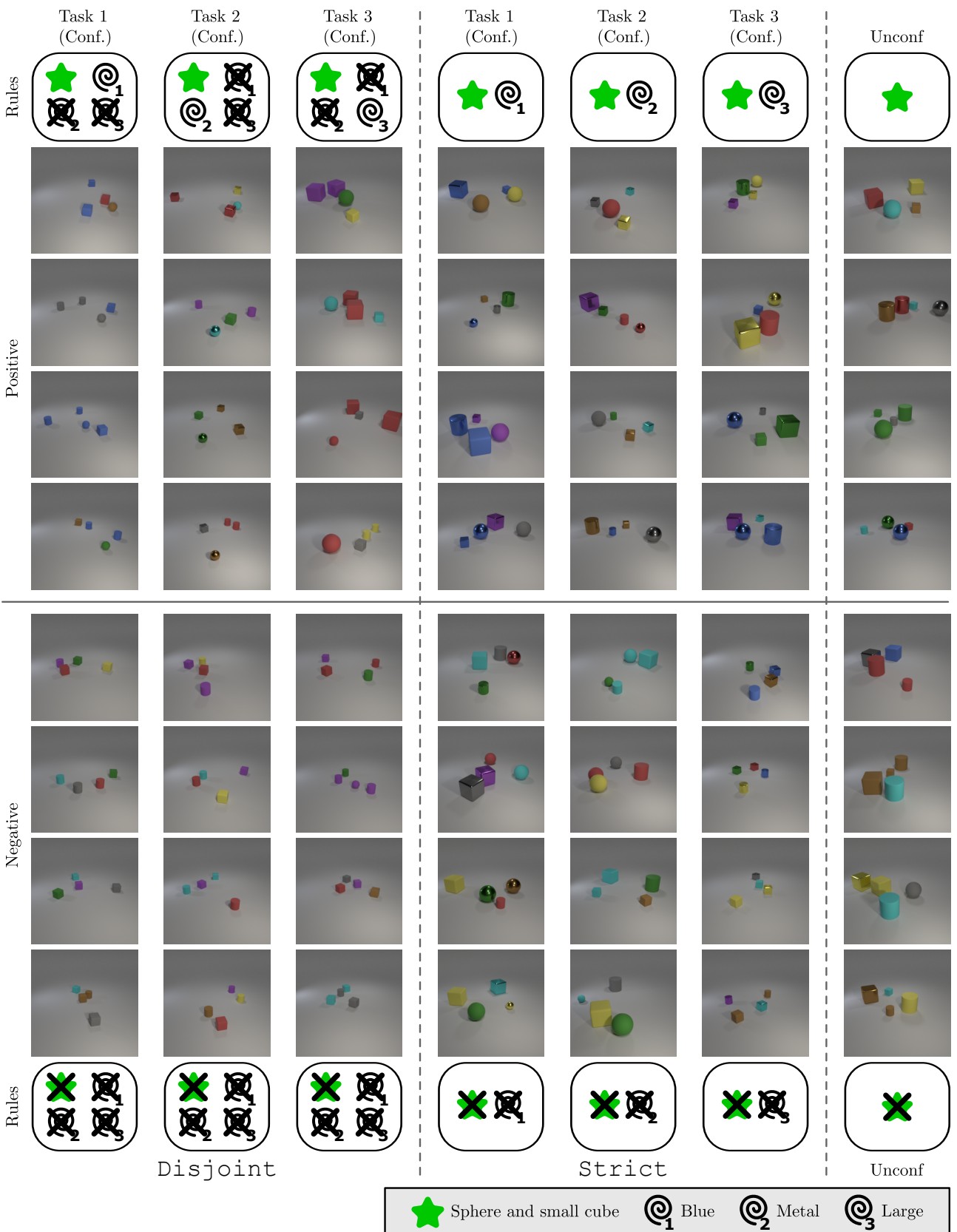

Figure 4. **Samples for the ConCon Dataset.** The first four samples for each dataset variant and task.

*Table 4.* **Rules `strict`:** a confounding feature must appear in the positive images but may not appear in the negative images of its respective task. The confounding features may, but are not required to, appear in images of other tasks.

|  | Positive | Negative |
|---|---|---|
| Task 1 | $g \wedge c_1$ | $\neg g \wedge \neg c_1$ |
| Task 2 | $g \wedge c_2$ | $\neg g \wedge \neg c_2$ |
| Task 3 | $g \wedge c_3$ | $\neg g \wedge \neg c_3$ |

Note that the intended behavior for our data generation consists of using simple confounders that are only required to be satisfied on a single object. Here, our code could easily be adapted, for example, to support the generation of images based on multi-object confounders by changing the selection of object properties in the respective function. In a similar spirit, both CLEVR and ConCon are generated with the Blender graphics simulation software, allowing for the inclusion of entirely new attributes and confounding features themselves. As such, extensions to, *e.g.*, novel color choices are realizable with either trivial or modest amounts of specification/change in code, or other already available properties (such as relationships) could easily be leveraged in future dataset creation and empirical investigation. Whereas these even more challenging setups may certainly be of interest in the long term, we note that our presently chosen configuration already highlights the catastrophic behavior of investigated methods and exposes insidious continual confounding, making solving our generated ConCon variant first desirable.

## B. Experimental Details

In this section, we give detailed information on the continual learning methods and the training hyperparameters used for ConCon training of NN and NeSy models, as well as a brief description of technical details.

**Methods.** Continual learning methods aim to alleviate forgetting when training and updating models continually. We here give additional information on the CL methods that we consider in our evaluation in the main paper: i) **EWC** (Kirkpatrick et al., 2017) is a method that regularizes weights relevant to the previous task, ii) **FROMP** (Pan et al., 2020) stores examples relevant to the previous task and performs functional regularization while training on the current tasks, iii) **ER** (Chaudhry et al., 2019) maintains a fixed-size memory buffer of random samples from prior tasks and interleaves them while training on the current task, iv) **DER** (Buzzega et al., 2020) also maintains a memory buffer that stores past samples and predictions, and in addition, the loss function includes a cross-entropy loss for the current task and a regularization term using the stored logits for past tasks, v) **BGS** (Lee et al., 2024) tackles the problem of spurious correlations in continual learning by storing samples that are balanced across the class labels and spurious features, vi) **GDUMB** (Prabhu et al., 2020) stores uniformly balanced past samples but is not a conventional continual learner as the model is trained from scratch at every new task, and vii) **PNNs** (Rusu et al., 2022) progressively grow the network by adding new subnetworks for each task, expanding the structure and parameters of previously learned networks.

**NN Training.** We use a ResNet-18 (He et al., 2016) model, a convolutional neural network-based architecture with skip connections. Following standard deep learning practice, the model is trained with an Adam optimizer (Kingma & Ba, 2014) with a mini-batch size of 64 and a learning rate of 0.001. The model was trained for 50 epochs for all methods but FROMP, where we trained the NN model for 30 epochs, as the performance became worse when training for 50.

**NeSy Training.** We use a neuro-symbolic concept learner (NeSy) (Stammer et al., 2021) that is equipped with a Slot attention model and a Set Transformer. The Slot-Attention module of NeSy is pre-trained on the original CLEVR dataset (Locatello et al., 2020) for the purpose of extracting accurate neuro-symbolic intermediate representations (*e.g.*, disentangled slots for color and shape). We use the (perfectly accurate as a consequence of simulation) structured information from the embeddings of the attention module to then learn only the Set Transformer for the ConCon dataset. The latter is trained using an Adam optimizer (Kingma & Ba, 2014) with a mini-batch size of 64 and a learning rate of 0.001. The model was trained for 150 epochs. We apply transformations to resize the images to $128 \times 128$ to fit the constraints set by prior works (Stammer et al., 2021).

**Hyperparameter Tuning.** We did not excessively tune hyperparameters as all models achieved good solutions when expected in their baseline scenario. As explained in the main body and discussed in much more detail in the next section, observed failure cases are systematic and structural in nature - a consequence of the continual learning set-up rather than hyperparameter values.

**Technical Details.** We created the dataset using an AMD EPYC 7742 64-Core Processor CPU. All models were built using PyTorch and trained and evaluated on an NVIDIA Tesla V100-SXM3-32GB with 32 GB of RAM. Training on a single task took no more than one hour for any experiment, with the exception of PNNs, which sometimes exceeded one hour.

## C. Further Analysis of Results

We now go into some further detail on our evaluation described in the main paper. First, we comment on standard deviations across experimental seeds, which have previously been omitted since they were negligible. We then showcase a set of training curves to corroborate our main body's results. Finally, additional empirical details for our main evaluation are shown. Here, we further explain the models' behaviors by considering accuracies on the decomposed sets of positive and negative images in addition to their average.

### C.1. Standard Deviations Across Experimental Repetitions with Random Seeds

In the main paper, we observed methods in the continual setting performing well on the tasks they were trained on but completely failing on the unconfounded dataset in the `disjoint` case and also not reaching the joint accuracy in the `strict` case. To facilitate visual interpretation and focus on the main insights, we omitted reporting of standard deviations arising from experimental repetition with five random seeds in the main body. We now provide more details on why we consider these deviations to be negligible overall, but in particular with respect to the drawn conclusions.

In Table 5, we see that GDUMB and FROMP are the two methods that generally have a larger standard deviation. GDUMB selects a small random buffer of samples to train on, ignoring the rest of the dataset. Hence, the data on which the model is trained differs among seeds. FROMP stores samples that are close to the decision boundary of previous tasks. In the `strict` case, if the ground truth is not learned and instead a decision boundary based on task-specific confounders is learned, then the selected samples are not helpful for learning the ground truth rule but, at the same time, a rule based on all confounders will not be able to classify images from all tasks correctly. As such, selecting samples close to the decision boundary may lead to unstable (*i.e.*, high standard deviation) results. Overall, the mean accuracies of GDUMB and FROMP are sufficiently low that the higher deviations do not affect the conclusions drawn.

Other than these two methods, standard deviations are smaller than 1 percentage point of accuracy on all current tasks except for the naive training on the NN, where one outlier causes a higher deviation, the third task for DER with a still small standard deviation of 2.69, and for the shuffled and cumulative settings, where it goes up to 3.85. For the latter two, the higher deviation is probably due to the model being trained on equal amounts of data from different tasks at the same time, therefore not being able to prioritize single tasks as much as other tasks, which almost reach perfect accuracy and thus have smaller deviations. We observe stronger deviations for the naive setup as well as EWC for old tasks. This fits with our observations of both approaches not successfully combating forgetting, resulting in partly random behavior on previous tasks. While there are some other slightly higher standard deviations of ER, DER, and BGS on old tasks as well, the largest deviation of these here is 5.79, which is still not relevant to any claims in our paper as the mean performances here are all below 60 and deviations as implied by the standard deviation still results in bad accuracies.

Most importantly, standard deviations on the unconfounded datasets are fairly small throughout. With the exception of the shuffled and cumulative settings, we encounter the largest standard deviation for the NN with FROMP at around 2.77, with all others being smaller. This highlights the consistency of the failure to learn the ground truth rule. On the shuffled training setup, standard deviation reaches 4.63 (less than 1 for NeSy) and on the cumulative training setup, standard deviation on the third task moves up to 5.78 for NN (less than 1 for NeSy). Whereas this is, in fact, a large and relevant deviation from the mean, it is still a big difference from the more than 20 points difference that is observed by comparing the joint and the cumulative settings or the more than 15 points difference between shuffled and cumulative. This further emphasizes the NN's preference for learning confounders over the ground truth in continual settings.

### C.2. Training Converges on Confounded Data with Worse Results on Unconfounded Data

Recall that, in the `strict` setting, we saw the models and the NN in particular achieve a high accuracy on the unconfounded dataset in the joint setting, while training on the same data in a cumulative manner resulted in an accuracy more than 20 points worse. This result is surprising since both setups are trained on the exact same datasets. Hence, one might wonder whether the model actually converged to a good accuracy during training in the cumulative setup. As we can see in Figure 5, both models indeed converge at close to perfect accuracy. For example, the NN ends at 100% train accuracy after task 1 and

*Table 5.* **ConCon Evaluation with Standard Deviations.** Test accuracies on `disjoint` and `strict` for NN and NeSy, including standard deviations. Coloring is white until a standard deviation of 1, and then takes darker shades of purple until 10. With a few exceptions, most standard deviations are relatively small. Most importantly, performance on the unconfounded data is consistently bad.

| | | Method | Current Task Acc. | | | Old Task Acc. | | Unconf. |
|---|---|---|---|---|---|---|---|---|
| | | | $T_1$ | $T_2$ | $T_3$ | $T_1$@$T_3$ | $T_2$@$T_3$ | |
| disjoint | NN | Naive Sequential | 100.0±0.00 | 92.2±15.58 | 100.0±0.00 | 50.0±0.03 | 63.6±18.02 | 48.1±1.81 |
| | | EWC (Kirkpatrick et al., 2017) | 100.0±0.00 | 99.9±0.07 | 100.0±0.00 | 50.1±0.21 | 60.6±13.97 | 47.7±2.18 |
| | | FROMP (Pan et al., 2020) | 100.0±0.00 | 99.8±0.14 | 99.4±1.00 | 99.2±1.53 | 97.9±2.41 | 50.2±0.17 |
| | | ER (Chaudhry et al., 2019) | 100.0±0.00 | 99.6±0.36 | 100.0±0.03 | 100.0±0.00 | 99.6±0.22 | 50.1±0.10 |
| | | DER (Buzzega et al., 2020) | 100.0±0.00 | 99.9±0.07 | 100.0±0.03 | 99.2±1.31 | 98.1±1.47 | 50.2±0.08 |
| | | BGS (Lee et al., 2024) | 100.0±0.00 | 99.9±0.05 | 100.0±0.03 | 95.1±4.08 | 96.2±2.03 | 50.4±0.20 |
| | | GDUMB (Prabhu et al., 2020) | 64.1±18.38 | 62.0±9.70 | 69.5±9.51 | 63.6±10.71 | 61.2±8.93 | 48.8±0.76 |
| | | PNNs (Rusu et al., 2022) | 100.0±0.00 | 99.8±0.05 | 100.0±0.00 | 100.0±0.00 | 99.8±0.05 | 47.9±2.36 |
| | | **Shuffled** | **100.0±0.03** | **99.9±0.03** | **100.0±0.03** | **100.0±0.03** | **99.8±0.05** | **50.0±0.03** |
| | | **Cumulative** | **100.0±0.00** | **100.0±0.03** | **100.0±0.03** | **100.0±0.05** | **99.9±0.08** | **49.9±0.00** |
| | | **Joint** | **100.0±0.00** | **99.9±0.04** | **100.0±0.03** | **N/A** | **N/A** | **50.0±0.03** |
| | NeSy | Naive Sequential | 99.5±0.18 | 99.9±0.12 | 100.0±0.03 | 50.0±0.00 | 58.1±4.81 | 50.0±0.07 |
| | | EWC (Kirkpatrick et al., 2017) | 99.6±0.28 | 100.0±0.03 | 99.9±0.05 | 50.2±0.35 | 67.2±9.12 | 50.2±0.18 |
| | | FROMP (Pan et al., 2020) | 99.6±0.31 | 98.9±0.29 | 98.4±0.69 | 97.9±1.45 | 97.3±0.72 | 50.5±0.23 |
| | | ER (Chaudhry et al., 2019) | 99.5±0.17 | 99.8±0.05 | 99.9±0.04 | 99.2±0.19 | 99.3±0.23 | 50.7±0.18 |
| | | DER (Buzzega et al., 2020) | 99.6±0.22 | 99.7±0.13 | 99.9±0.08 | 98.6±0.64 | 96.9±1.12 | 50.9±0.05 |
| | | BGS (Lee et al., 2024) | 99.6±0.20 | 99.9±0.09 | 99.9±0.05 | 94.4±2.52 | 95.1±1.78 | 50.9±0.19 |
| | | GDUMB (Prabhu et al., 2020) | 99.4±0.12 | 97.8±0.89 | 98.5±0.72 | 98.0±0.86 | 98.2±0.90 | 50.3±0.15 |
| | | PNNs (Rusu et al., 2022) | 99.5±0.25 | 100.0±0.00 | 100.0±0.00 | 99.6±0.27 | 100.0±0.00 | 49.6±0.23 |
| | | **Shuffled** | **99.8±0.09** | **99.7±0.09** | **99.9±0.05** | **99.7±0.13** | **99.9±0.03** | **50.8±0.07** |
| | | **Cumulative** | **99.8±0.07** | **99.9±0.09** | **99.9±0.10** | **99.8±0.07** | **99.9±0.03** | **50.8±0.11** |
| | | **Joint** | **99.7±0.10** | **99.9±0.05** | **99.9±0.08** | **N/A** | **N/A** | **50.7±0.04** |
| strict | NN | Naive Sequential | 100.0±0.00 | 100.0±0.00 | 100.0±0.00 | 49.1±0.05 | 48.9±0.07 | 50.0±0.07 |
| | | EWC (Kirkpatrick et al., 2017) | 100.0±0.00 | 99.9±0.10 | 100.0±0.05 | 49.1±0.05 | 48.9±0.08 | 50.0±0.07 |
| | | FROMP (Pan et al., 2020) | 100.0±0.00 | 59.9±8.49 | 69.2±4.92 | 67.8±10.71 | 59.1±6.35 | 51.1±2.77 |
| | | ER (Chaudhry et al., 2019) | 100.0±0.00 | 97.7±0.50 | 96.4±0.59 | 54.6±1.76 | 59.6±2.71 | 51.0±1.24 |
| | | DER (Buzzega et al., 2020) | 100.0±0.00 | 98.7±0.92 | 96.5±2.69 | 52.8±4.13 | 55.5±5.79 | 49.1±0.26 |
| | | BGS (Lee et al., 2024) | 100.0±0.00 | 99.8±0.12 | 99.6±0.16 | 50.1±1.00 | 49.1±1.01 | 49.5±0.32 |
| | | GDUMB (Prabhu et al., 2020) | 92.5±2.77 | 62.7±8.34 | 53.6±2.32 | 57.5±5.20 | 54.2±2.89 | 50.4±0.50 |
| | | PNNs (Rusu et al., 2022) | 100.0±0.00 | 99.8±0.09 | 99.8±0.42 | 100.0±0.00 | 99.8±0.09 | 49.2±1.23 |
| | | **Shuffled** | **79.9±3.11** | **93.4±3.85** | **95.3±2.14** | **95.2±3.21** | **96.7±1.76** | **89.5±4.63** |
| | | **Cumulative** | **100.0±0.00** | **87.0±1.57** | **88.8±3.65** | **85.9±2.65** | **91.0±2.89** | **72.6±5.78** |
| | | **Joint** | **98.2±1.11** | **99.2±0.17** | **98.9±0.30** | **N/A** | **N/A** | **95.7±2.04** |
| | NeSy | Naive Sequential | 99.7±0.20 | 100.0±0.03 | 100.0±0.00 | 49.1±0.08 | 49.2±0.09 | 49.8±0.05 |
| | | EWC (Kirkpatrick et al., 2017) | 99.9±0.08 | 99.9±0.07 | 100.0±0.03 | 49.1±0.08 | 49.4±0.49 | 50.1±0.19 |
| | | FROMP (Pan et al., 2020) | 99.8±0.12 | 83.4±9.01 | 74.7±7.18 | 85.1±7.65 | 76.3±10.56 | 56.2±1.17 |
| | | ER (Chaudhry et al., 2019) | 99.8±0.17 | 99.8±0.15 | 98.8±0.11 | 68.0±1.42 | 83.6±0.71 | 61.2±0.82 |
| | | DER (Buzzega et al., 2020) | 99.7±0.23 | 99.4±0.15 | 99.5±0.24 | 58.8±1.30 | 73.2±3.91 | 56.5±1.05 |
| | | BGS (Lee et al., 2024) | 99.8±0.12 | 99.9±0.03 | 99.9±0.08 | 51.8±1.01 | 56.3±1.55 | 51.3±0.49 |
| | | GDUMB (Prabhu et al., 2020) | 99.4±0.24 | 85.6±1.39 | 82.7±1.64 | 80.1±2.42 | 84.2±2.33 | 56.7±1.09 |
| | | PNNs (Rusu et al., 2022) | 99.8±0.07 | 100.0±0.03 | 100.0±0.00 | 99.8±0.09 | 100.0±0.03 | 49.9±0.05 |
| | | **Shuffled** | **87.8±0.66** | **92.5±0.59** | **89.2±0.60** | **88.2±0.35** | **92.6±0.79** | **70.1±0.51** |
| | | **Cumulative** | **99.8±0.09** | **92.5±0.71** | **88.8±0.51** | **88.6±0.65** | **92.3±0.65** | **69.9±0.81** |
| | | **Joint** | **88.4±0.40** | **92.6±0.21** | **89.8±0.23** | **N/A** | **N/A** | **70.8±0.33** |

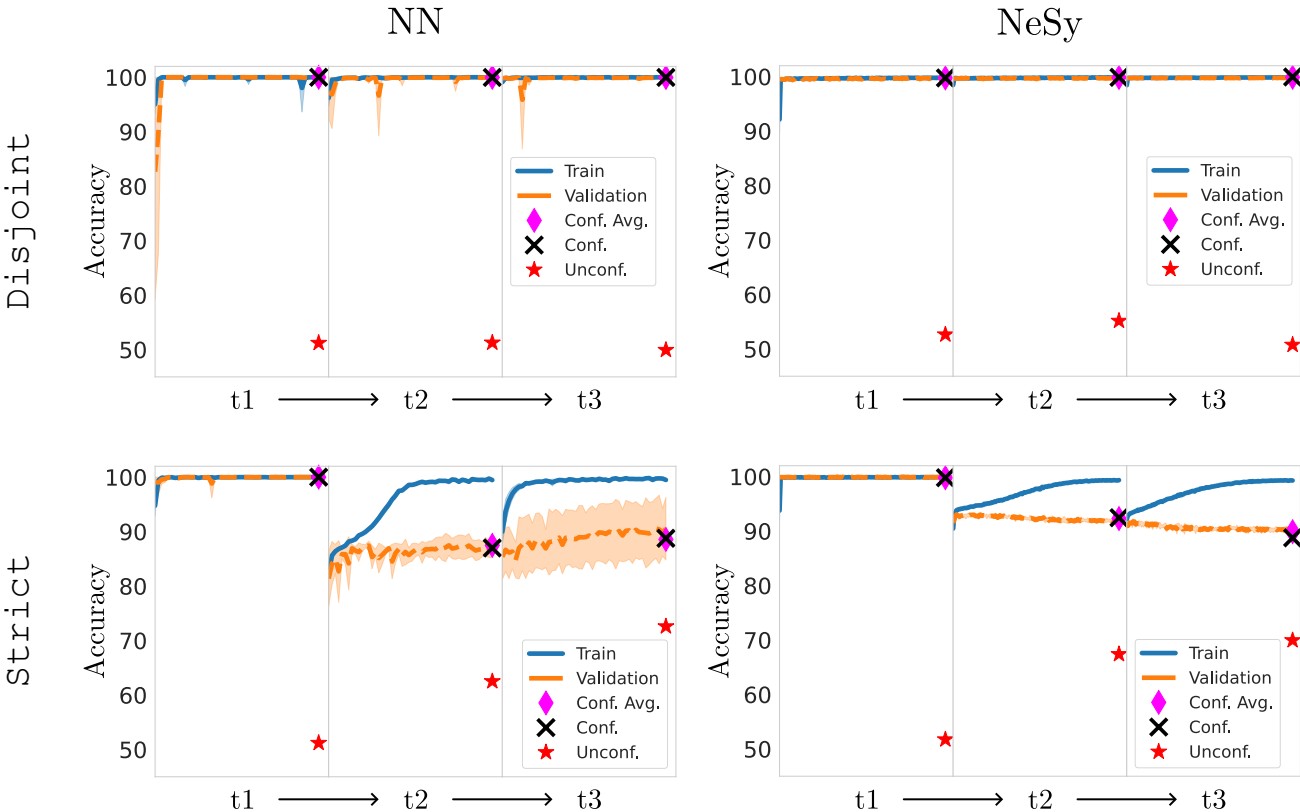

*Figure 5.* **Learning Curves for the Cumulative Training Setup.** Recall that in the cumulative setting, the dataset being trained on grows over time, *i.e.*, all datasets from previous tasks are concatenated with the currently observed one. This figure compares these curves to the test accuracy on the unconfounded dataset (denoted by Unconf.), task-specific confounded datasets (denoted by Conf.), and average accuracy calculated over accumulated confounded tasks (denoted by Conf. Avg.) at the end of the training. Average accuracy after $T$ tasks is given as: $A_T = \frac{1}{T} \sum_t^T a_t$. Observe that in the strict case, performance on the unconfounded data does increase during training, but the final performance is still poor. We expect that this can be explained by the further observation of the divergence between train and validation curves in the strict case.

99.5% accuracy after task 2. Training on the full accumulated data in task 3 peaks at 99.96% and ends at 99.68% accuracy. Despite this, performance on the unconfounded dataset, while increasing with the number of tasks, remains significantly worse than the train accuracy. In the main paper, we showed that the easiest rule that can be learned to solve all confounded tasks jointly is the ground truth rule. If that is true, why does the NN achieve a near-perfect train accuracy after training on all tasks but fail on the unconfounded data where the ground truth also applies? By inspecting the validation and test values of the confounded datasets, we find that, despite coming from the same distribution of images, the accuracies on these are considerably worse than for the train images. It turns out that the cumulative setting has resulted in both models overfitting far more severely to the train distribution. It thus underperforms on both confounded data from the same distribution and unconfounded data due to a general failure to fully generalize to new images. We expect that more investigation of this overfitting will further illuminate the problem of *insidious continual confounding*.

### C.3. Models Mostly Fail on Positive Images in `disjoint` Settings

Our main body's evaluation on the `disjoint` dataset showed that performance on the unconfounded dataset is not much better than random guessing, even when forgetting is successfully avoided. In this part of the appendix, we go into further detail on our evaluation to not only examine the average accuracies, but also the models' performance on the positive and negative images individually. Through this analysis, we can then pinpoint the failure modes even more precisely.

We employ a new visualization, extending the purely tabular results presented before. Here, we measure the accuracy on the confounded test dataset across all the tasks after training the model on each task. To investigate whether and how the model

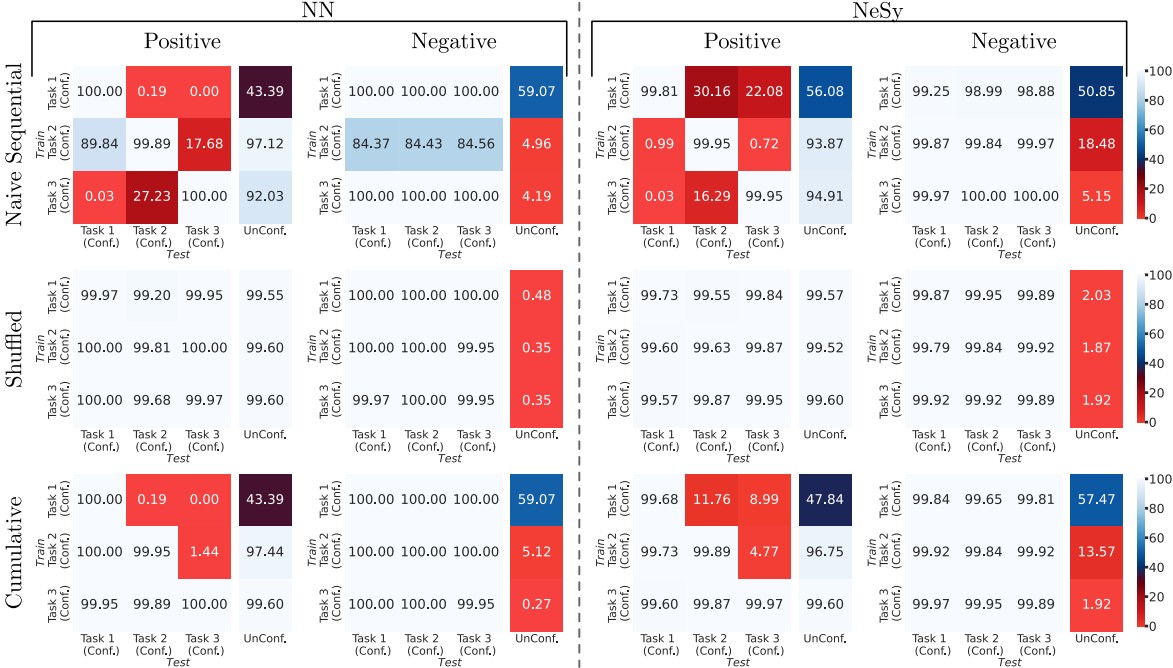

*Figure 6.* **Accuracies for Positive and Negative Images on `disjoint`.** Test accuracies on both the NN and NeSy for Naive Sequential, Shuffled, and Cumulative. We can see that the prediction error is a result of positive images being incorrectly classified as negative. This is consistent with our hypothesis that the model mostly bases its predictions on the presence of the confounding features, none of which appear in any negative images of the confounded tasks. On the unconfounded data, the high performances on positive images after tasks 2 and 3 indicate that a disjunction of confounding features is learned since this is satisfied in most positive and only a few negative images. Further results on the other continual learning methods can be found in Figure 7. (Best viewed in color)

gets confounded, we also measure the accuracy on the unconfounded dataset. These results are illustrated via an inter-task confusion matrix in Figures 6 and 7. Each row shows the results for a continual learning setup on both NN (left) and NeSy (right). Instead of the average accuracies already discussed in the main paper, we here plot the results on the positive and negative sets of images. We additionally color the fields corresponding to their accuracy, going from incorrect predictions (0%; red) over random predictions (50%; blue) to correct predictions (100%, white).

First, consider the shuffled approach. Here, tasks are not aligned with confounders, which is why performance on all tasks is very similar. Our hypothesis of the model learning a disjunction of confounders fits the results, as all tasks are predicted correctly but most negative images of the unconfounded datasets are incorrectly classified as positive. In the following, we go into further detail on the continual learning setups that are affected by confounders to further explain the models' behavior there.

The results of GDUMB on NN set themselves apart from the other methods. We will quickly cover the other methods and then return to GDUMB. With the exception of GDUMB, predictions on the negative set across all tasks and methods are mostly or even fully correct, and the classification errors are almost exclusively concentrated on the positive set. In other words, the models seem to predict the negative class too often, hence failing on the positive set. This can be explained by the model predicting the positive class whenever the confounding feature is present (it is never present on negative images of confounded tasks). The model also performs well on all previous tasks in FROMP, ER, DER, BGS, PNNs, and the cumulative setting, indicating that some combination of confounders is learned.

We can get some more insights into the model behavior by relating it to the performance on the unconfounded dataset. For all methods, performance on both positive and negative images is around 50% after the first task. Here, the model appears to make predictions based on the first confounding feature, *blue*, as in the unconfounded dataset, this roughly matches the proportion of images with and without a blue object. However, almost all images are predicted to be positive for the two following tasks. Our results on the memory-based methods FROMP, ER, DER, BGS, GDUMB, as well as PNNS and the cumulative setup, which show good performance on all current and previously seen tasks, lead us to hypothesize that the

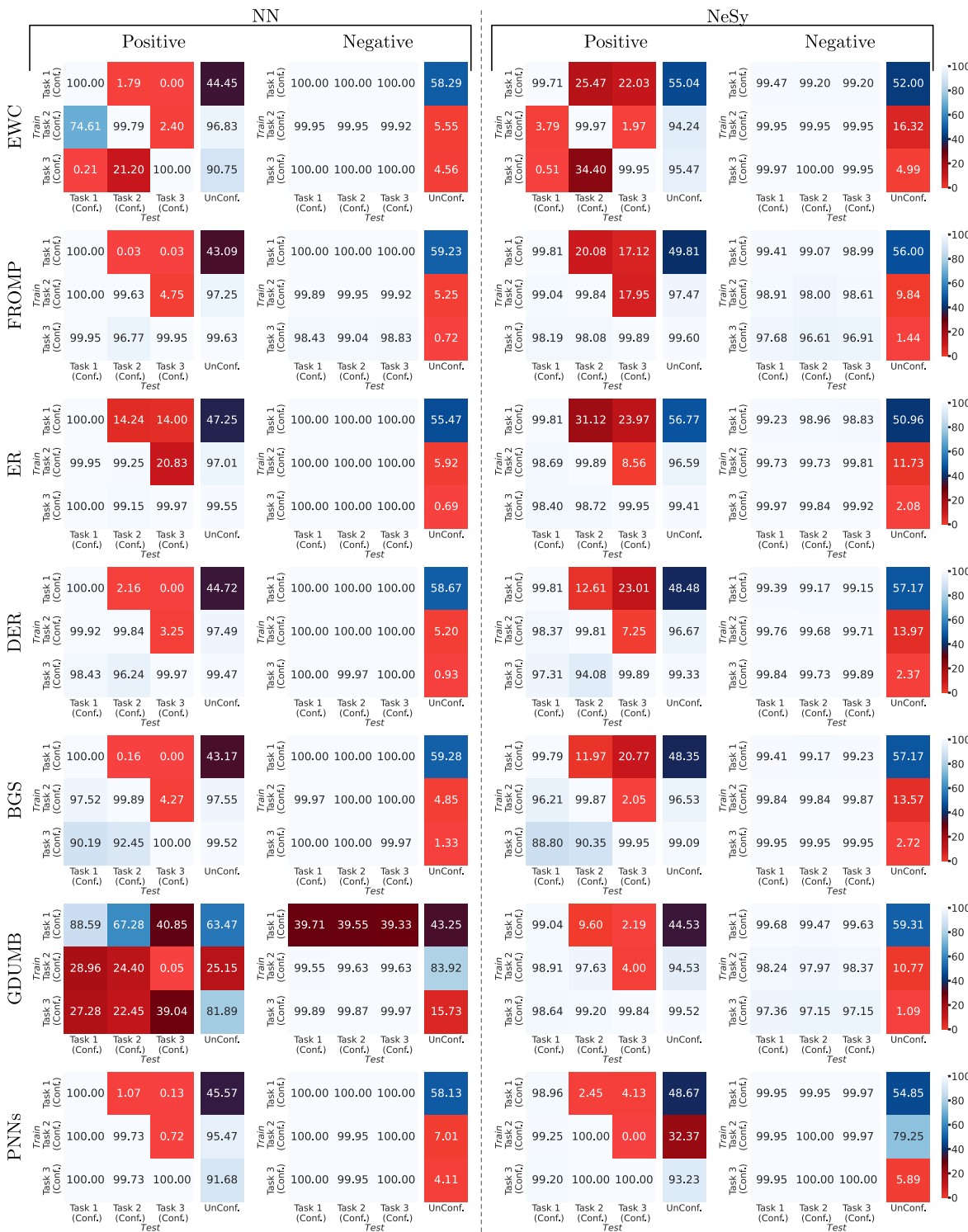

*Figure 7.* **Accuracies for Positive and Negative Images on `disjoint`.** Test accuracies on both the NN and NeSy for different continual learning methods. As for the baseline methods, we see that the model failures on confounded tasks are due to incorrectly predicted positive images, while too many unconfounded images are incorrectly predicted as belonging to the positive class. (Best viewed in color)

disjunction of all confounding features is learned. In any image of the unconfounded dataset, it is very likely that at least one confounding feature is present, hence resulting in predictions for the positive class. For the naive setup and EWC, the higher accuracies on the positive images can also partly be explained by the data generation process: for both *metal* and *large*, it is more likely for an unconfounded image to contain the respective features than not to contain them.

Now we return to GDUMB, which, training on only the buffer, learns something more random than the other methods (also see Table 5), resulting in more unpredictable results after training on task 1. After task 2, GDUMB follows similar patterns to the other methods but with worse results. GDUMB fails to learn the ground-truth rule and also does not learn a solution that makes correct predictions on all confounded tasks (such as the disjunction).

With respect to NeSy, the same patterns can be observed. Performance on task 1 after training on task 2 is worse than for the NN, possibly indicating a stronger focus on the confounder of the current task. We believe that the marginally better accuracies on some not-yet-seen tasks (forward transfer) can partially be explained by the model learning some aspect of the ground truth rule due to its neurosymbolic representation at the core.

Overall, results on the positive and negative images show that the negative set of confounded tasks can be consistently predicted correctly. This is entirely plausible and perhaps even expected, as the negative set used in all tasks follows shared rules for generation. In turn, errors are introduced by incorrect predictions on positive images. The accuracies on the unconfounded dataset conform with our hypothesis of the models learning a disjunction of confounders, a rule that is mostly satisfied on positive unconfounded images.

### C.4. Models Base Predictions on Task-Specific Confounders in `strict` Settings

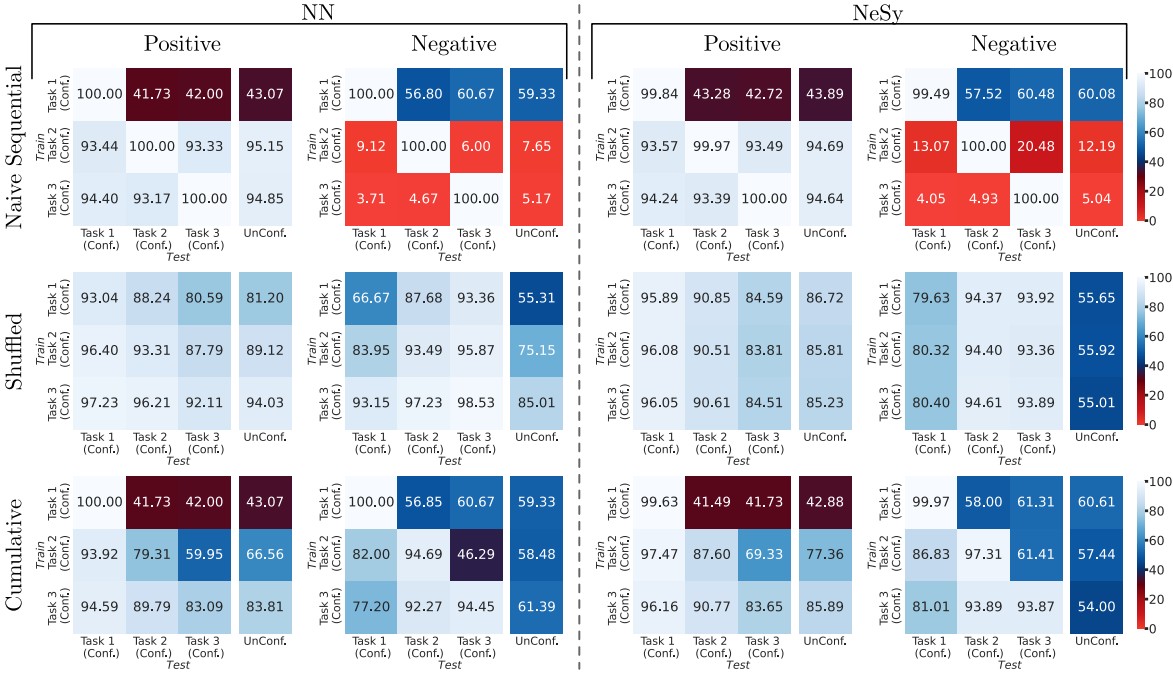

*Figure 8.* **Accuracies for Positive and Negative Images on `strict`.** Test accuracies on both the NN and NeSy for Naive Sequential, Shuffled, and Cumulative. For naive sequential, performance on positive and negative images is relatively balanced after task 1. For tasks 2 and 3, the respective confounding features are more likely to be present, so the model tends to classify images as "positive" in the naive sequential approach. This is due to the first confounding feature (*blue*) having more alternative values (there are eight colors) than the confounding features *metal* and *large*, which both have only one alternative property value. In the shuffled setup for NN, the accuracies are similar across tasks, with only the unconfounded accuracy improving with additional data. Cumulative also improves with additional data, but stays behind shuffled in terms of accuracy. (Best viewed in color)

For the `strict` dataset, our evaluation in the main paper demonstrated the problematic manifestation of *insidious continual confounding*, *i.e.* a model's performance when trained on all confounded tasks jointly is considerably higher than when trained on the same data in a cumulative, continual setup. In the latter, it largely fails.

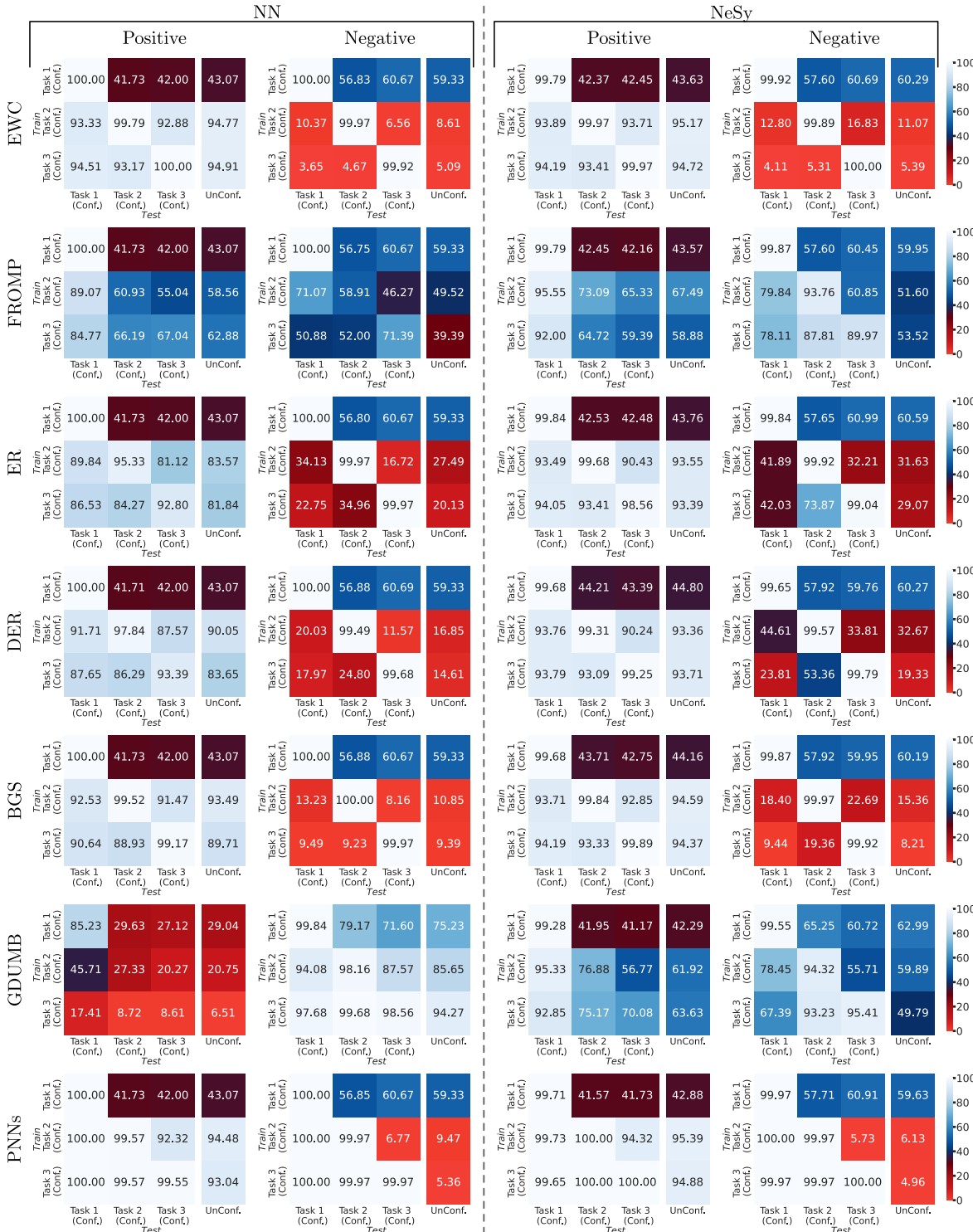

*Figure 9.* **Accuracies for Positive and Negative Images on `strict`.** Test accuracies on both the NN and NeSy for different continual learning methods. Similarly to naive sequential, performance on positive and negative images is mostly balanced after task 1. In the second and third tasks, the models tend to classify images as "positive" more often, as confounding features are more likely to be present than not. (Best viewed in color)

Figures 8 and 9 show the individually decomposed results for the `strict` dataset variant, analogous to the analysis of the previous subsection. As for the `disjoint` case, accuracies in the shuffled case are mostly unchanged throughout tasks because the tasks are not aligned with the confounding features. We can, however, see that additional data helps the model learn the ground truth rule, as accuracies on both positive and negative images go up in tasks 2 and 3. Considering the continual learning setups, we again see that the models appear to have learned the first confounder after the first task. However, due to this confounding feature being allowed to be present in other tasks, this now leads to more balanced predictions on future tasks. After training on tasks 2 and 3, most predictions fail on the negative instead of the positive sets of images. This is again consistent with our hypothesis of the model learning the task-specific confounder. To be precise, both the confounders of tasks 2 and 3 describe a property, which, in our dataset, only has two different possible values (*metal* or *rubber*; *small* or *large*). Since these properties are determined randomly if not specified, there is only a small probability of that feature not being present.[8] Therefore, applying the rule based on the task-specific confounding feature mostly classifies any such image as positive, no matter the class, thus adding up to around 50% on average.

While the results for the CL methods look mostly similar, there are three methods with results different from the others. FROMP has worse results on positive but better results on negative images. Here, old task accuracy is improved at the cost of current task performance. While this indicates that something different from simply the current task confounder is learned, the unconfounded accuracy remains to be not much higher than 50%. The second outlier method, GDUMB, performs better on negative images but much worse on positive ones. Since, just like with the cumulative training, the ratio of samples from each class is the same, it does not prioritize the current task confounder. However, it still fails to make correct predictions on many images of the confounded tasks and also gives bad predictions on the unconfounded dataset. Lastly, PNNs mostly fall in line with the other methods, but the task information allows them to correctly predict positive and negative images of all previous tasks. On the other hand, this still does not help with making correct predictions on the unconfounded data, where PNNs do not perform better than the other methods.

NeSy can be observed to perform less badly on negative images. It is possible that it picks up on some parts of the ground truth rule more successfully than the NN. Nevertheless, as we have seen in the main paper, the overall performance on the unconfounded dataset still has a large margin of error.

In summary, the models mostly focus on the task-specific confounders in the `strict` case. Whether such a model performs better on one class or the other is then determined by the distribution of test images. In this `strict` variant of ConCon, confounding features are likely to be present on any image of other tasks and the unconfounded dataset, thus resulting in better predictions on positive and worse predictions on negative images.

## D. Experiment Using Generative Models

*Table 6.* **Test Accuracies for Generative Replay on the ConCon Dataset**. While accuracies on the current tasks are high, we observe catastrophic forgetting, resulting in low accuracies on old tasks. Accuracy on the unconfounded dataset is around $50\%$, showing that the model does not successfully learn the ground truth rule.

|          | $T_1$ | $T_2$ | $T_3$ | $T_1@T_3$ | $T_2@T_3$ | **UnConf** |
|---|---|---|---|---|---|---|
| disjoint | 100.0±0.0 | 88.78±14.29 | 99.28±1.02 | 53.2±6.30 | 59.33±15.02 | 47.73±1.45 |
| strict | 99.96±0.08 | 98.54±0.51 | 99.58±0.46 | 48.84±0.48 | 48.61±0.56 | 49.26±0.88 |

In addition to the NN and NeSy models, we here want to investigate whether another type of model based on generative models is more successful in overcoming continual confounding. To this end, we use a Variational Autoencoder (VAE) architecture based on ResNet-18 following van de Ven et al. (2020). The VAE framework allows the model to generate synthetic samples of the ConCon dataset from previous tasks by sampling from the learned latent space, providing a mechanism for replay without storing actual historical data. This experiment uses the same hyperparameters as the experiment in the main body (see Appendix B). Table 6 shows the experimental results on both the `disjoint` and `strict` variants of our ConCon dataset. The model achieves high accuracies on the current tasks across both `disjoint` and `strict` settings, but fails at correctly predicting images from previous tasks, showcasing catastrophic forgetting. More critically (but, given the bad performance on previous tasks, also unsurprisingly), the model fails completely on the

---

[8]Each of the four objects in an unconfounded image has a chance of containing the material *metal*. The probability of such an image not containing any *metal* object is, therefore, quite low. The same idea applies analogously to *large* (on positive images, the *small sphere* part of the ground truth rule also affects the distribution of sizes).

unconfounded dataset, indicating, like other continual learning methods, that it learns the task-specific confounders and not the ground truth rule.

## E. Expanded ImageNet Experiments

Following up on our motivational experiment using ImageNet classes at the start of the main body, we now conduct a larger experiment using ImageNet classes to further investigate whether we again observe a gap between joint and cumulative training. To this end, we consider two categories: four-legged animals and non-four-legged animals. We define three tasks where each task contains data from eight ImageNet classes, half of which are four-legged animals. Additionally, in each task, this group of four ImageNet classes shares a similarity with respect to color. For example, the two resulting classes in task 1 contain either four-legged animals where the predominant color is white (*e.g.*, arctic fox, polar bear), while the second class contains four-legged animals where the predominant color is blue (*e.g.*, jellyfish, blue jay). Therefore, when training the model in a continual manner (cumulative training), it is likely that the model will start by learning a simpler rule in task one, by, for example, making predictions mostly based on color, the "four-legged" feature, or even other features that we did not consider. Following our insights from the experiments of the main body, we hypothesize that such behavior will lead to the model having more trouble with correctly making predictions on all data compared to the joint setting, where it processes all the data at the same time and, hence, would not favor rules as much that are only consistent with a subset of all classes.

The exact description of the dataset and experimental results is shown in Table 7. We use a ResNet-18 model and average results across 5 random seeds (other hyperparameters are the same as for most other experiments, see Appendix B). The table shows the accuracies after the model has been trained on all 3 tasks. From joint training to cumulative training, we observe an average decrease in accuracy of 5.58 percentage points, a substantial degradation in performance. While the setup is slightly different from our continual confounding experiments in the main paper, it does showcase that the continual learning setting tends to result in a less accurate model than the joint setup.

*Table 7.* **Cumulative and Joint Training on ImageNet Data.** At the top, we show what ImageNet classes the dataset consists of. Each task includes a pair of classes defined by "4-leggedness" and dominant color, resulting in six classes overall. Class examples are from ImageNet. At the bottom, we report cumulative and joint training accuracies after training on all tasks. On average, joint training has an accuracy of 5.58 points higher than cumulative training.

| Task | Class | Description | ImageNet Classes |
|------|-------|-------------|------------------|
| $T_1$ | $C_0$ | Four-legged, white color | arctic fox, polar bear, white wolf, samoyed dog |
|       | $C_1$ | Not four-legged, blue color | jellyfish, blue jay, blue shark, tench |
| $T_2$ | $C_2$ | Four-legged, green color | cheetah, deer, lion, leopard |
|       | $C_3$ | Not four-legged, white color | snowmobile, sailboat, snowplough, rattlesnake |
| $T_3$ | $C_4$ | Four-legged, blue color | hippo, crocodile, water buffalo, beaver |
|       | $C_5$ | Not four-legged, other colors | eagle, spider, lifeboat, mushroom |

| **Accuracy After Training on All Tasks** | | | |
|------------------|-----------|-----------|-----------|
| **Training Setting** | $T_1$ | $T_2$ | $T_3$ |
| Cumulative | 76.30±0.45 | 78.50±2.69 | 84.40±0.86 |
| Joint | 83.45±2.46 | 86.25±1.24 | 86.25±1.15 |

## F. Limitations

ConCon is the first dataset designed to investigate how ML and CL methods behave in the case of continual confounding. Nevertheless, there are certain possible research directions on continual confounding variations where an extension of the current ConCon could be beneficial. While our code framework for generating datasets can be changed to allow for different kinds of continual confounding, the current implementation only aims to generate problems where confounders only appear on a single object. Multi-object confounders are not implemented as of now. Other types of confounders, such as relations between objects or positioning (*e.g.*, *there is a sphere left to a red object*, *there is a cube on the far left*), are also not considered yet. Generally, any CLEVR-based dataset will have limitations due to its synthetic nature. Investigating continually confounded real-world datasets thoroughly is thus left for future work.

## G. ConCon Datasheet

| Motivation |
| --- |

**For what purpose was the dataset created?** (*e.g.*was there a specific task in mind? Was there a specific gap that needed to be filled? Please provide a description.)

The ConCon dataset was created to evaluate confounding in a continual learning setting. It consists of a learning task which can be solved by applying a ground truth rule. There exists one unconfounded dataset where no confounder is present and the ground truth determines the class assignment. Additionally, there are two dataset variants, `disjoint` and `strict`, with three confounded tasks each. Each task consists of its own dataset. In a continual learning setting, a model is only trained on data from a single task at the same time, going over all tasks in sequential order. For `disjoint`, the confounding features are only present in their respective tasks. For `strict`, the confounding features are only informative in their respective tasks, *i.e.* they may only appear in one class of the confounded task but in any class of the other tasks.

**What (other) tasks could the dataset be used for?** Are there obvious tasks for which it should not be used?

The primary purpose of ConCon is the investigation of *continual confounding*. It could also be used for other tasks, such as neuro-symbolic continual learning.

**Who created this dataset (e.g., which team, research group) and on behalf of which entity ?** (e.g., company, institution, organization)

 The creators of the dataset are: Florian Peter Busch[1,2], Roshni Kamath[1,2], Rupert Mitchell[1,2], Wolfgang Stammer[1,2], Kristian Kersting[1,2,3,4] and Martin Mundt[5]. The associated affiliations are: [1]Department of Computer Science, TU Darmstadt, Darmstadt, Germany, [2]Hessian Center for AI (hessian.AI), Darmstadt, Germany, [3]German Research Center for Artificial Intelligence (DFKI), Darmstadt, Germany and [4]Centre for Cognitive Science, TU Darmstadt, Darmstadt, Germany. [5]University of Bremen, Germany.

**Who funded the creation of the dataset?** If there is an associated grant, please provide the name of the grantor and the grant name and number.

No funding was received for the creation of the dataset.

**Any other comments?**

No further comments.

| Composition |
| --- |

**What do the instances that comprise the dataset represent (e.g., documents, photos, people, countries)?** Are there multiple types of instances (e.g., movies, users, and ratings; people and interactions between them; nodes and edges)? Please provide a description.

Each instance is an image representing four geometric objects with certain properties. An object can have any of three shapes (*cube*, *sphere*, *cylinder*), be either *small* or *large*, have either *metal* or *rubber* material and be one of eight colors (*gray*, *red*, *blue*, *green*, *brown*, *purple*, *cyan*, *yellow*).

**How many instances are there in total (of each type, if appropriate)?**

Each task consists of 9,000 instances. There is one unconfounded dataset and both the `disjoint` and `strict` variants consist of three tasks each - resulting in a total of 63,000 instances.

**Does the dataset contain all possible instances or is it a sample (not necessarily random) of instances from a larger set?** If the dataset is a sample, then what is the larger set? Is the sample representative of the larger set (e.g., geographic coverage)? If so, please describe how this representativeness was validated/verified. If it is not representative of the larger set, please describe why not (e.g., to cover a more diverse range of instances, because instances were withheld or unavailable).

The dataset was generated synthetically. The entire space of possible samples is given by any combination of properties of the four objects included in an image and their placement. Additionally, images which do not conform to the dataset setup as determined by the ground truth rule and confounding features are not sampled. The dataset was sampled from the entire space of such permissible instances. The positioning of objects in an image is also determined randomly, avoiding images where an object is not visible (*e.g.* an object is hidden behind another).

**What data does each instance consist of? "Raw" data (e.g., unprocessed text or images) or features?** In either case, please provide a description.

Each instance consists of a raw image in png format showing the objects as described previously.

**Is there a label or target associated with each instance?** If so, please provide a description.

Each instance fulfills or does not fulfill a (ground truth) rule. Thus, instances could be seen as belonging to a *positive* or *negative* class, and are provided already separated into these categories.

**Is any information missing from individual instances?** If so, please provide a description, explaining why this information is missing (e.g., because it was unavailable). This does not include intentionally removed information, but might include, e.g., redacted text.

No information is missing that is relevant to the intents and purposes of the dataset's creation. We follow the typical omission of nuanced graphics engine simulation parameters.

**Are relationships between individual instances made explicit (e.g., users' movie ratings, social network links)?** If so, please describe how these relationships are made explicit.

Apart from sharing the rule determining the associated label, the dataset is constructed to be confounded. In `disjoint`, there is a confounding feature that is only present in the positive images of one specific tasks and does not appear anywhere else in `disjoint`. In `strict`, these confounding features are also present in the respective positive images and not in the same task's negative images but they may appear randomly in other tasks. The unconfounded dataset is not confounded. No further relationships are made explicit.

**Are there recommended data splits (e.g., training, development/validation, testing)?** If so, please provide a description of these splits, explaining the rationale behind them.

Yes, each task consists of a training, validation and testing data split, with 3000, 750, and 750 instances, respectively. Additionally, we also have a universal unconfounded dataset with the same split as the tasks. There is no difference in the data generation of these splits.

**Are there any errors, sources of noise, or redundancies in the dataset?** If so, please provide a description.

There are no errors and no noise other that which is is implicitly caused by the simulation process. There is a limited set of configurations resulting in an instance. Nevertheless, this set is large but redundancy may occur. Note, that even if two data items have the same object properties, they are vanishingly unlikely to also have the same positions.

**Is the dataset self-contained, or does it link to or otherwise rely on external resources (e.g., websites, tweets, other datasets)?** If it links to or relies on external resources, a) are there guarantees that they will exist, and remain constant, over time; b) are there official archival versions of the complete dataset (i.e., including the external resources as they existed at the time the dataset was created); c) are there any restrictions (e.g., licenses, fees) associated with any of the external resources that might apply to a future user? Please provide descriptions of all external resources and any restrictions associated with them, as well as links or other access points, as appropriate.

The dataset is self-contained, however it does build on prior work. It is built on the CLEVR framework (Johnson et al., 2017), which has an open source license that allows us to make these modifications. Data generation also relies on the publicly accessible and free Blender software.

**Does the dataset contain data that might be considered confidential (e.g., data that is protected by legal**

**privilege or by doctor-patient confidentiality, data that includes the content of individuals non-public communications)?** If so, please provide a description.

No.

**Does the dataset contain data that, if viewed directly, might be offensive, insulting, threatening, or might otherwise cause anxiety?** If so, please describe why.

No.

**Does the dataset relate to people?** If not, you may skip the remaining questions in this section.

No, the dataset only contains synthetic objects.

**Does the dataset identify any subpopulations (e.g., by age, gender)?** If so, please describe how these subpopulations are identified and provide a description of their respective distributions within the dataset.

No.

**Is it possible to identify individuals (i.e., one or more natural persons), either directly or indirectly (i.e., in combination with other data) from the dataset?** If so, please describe how.

No.

**Does the dataset contain data that might be considered sensitive in any way (e.g., data that reveals racial or ethnic origins, sexual orientations, religious beliefs, political opinions or union memberships, or locations; financial or health data; biometric or genetic data; forms of government identification, such as social security numbers; criminal history)?** If so, please provide a description.

No.

**Any other comments?**

No further comments.

---

**Collection Process**

---

**How was the data associated with each instance acquired?** Was the data directly observable (e.g., raw text, movie ratings), reported by subjects (e.g., survey responses), or indirectly inferred/derived from other data (e.g., part-of-speech tags, model-based guesses for age or language)? If data was reported by subjects or indirectly inferred/derived from other data, was the data validated/verified? If so, please describe how.

This dataset was generated synthetically and by simulation according to certain rules as explained previously. Data was generated using Blender and not sampled from a real dataset.

**What mechanisms or procedures were used to collect the data (e.g., hardware apparatus or sensor, manual human curation, software program, software API)?** How were these mechanisms or procedures validated?

The source code used to generate the datasets is available at: `https://github.com/ml-research/concon`.

**If the dataset is a sample from a larger set, what was the sampling strategy (e.g., deterministic, probabilistic with specific sampling probabilities)?**

Sampling was conducted in a probabilistic manner where unassigned object properties are selected with a uniform probability. To ensure all objects are recognizable, images where only a low amount of an object's pixels are visible (for example due to occlusion) are rejected and replaced with a new sample. This generation procedure may thus result in non-uniform distributions in size and shape attributes.

**Who was involved in the data collection process (e.g., students, crowdworkers, contractors) and how were they compensated (e.g., how much were crowdworkers paid)?**

Code was written and data generation conducted by the authors of this dataset and its respective paper. No subcontractors or crowdworkers were involved as all data is of simulated nature.

**Over what timeframe was the data collected? Does this timeframe match the creation timeframe of the data associated with the instances (e.g., recent crawl of old news articles)?** If not, please describe the timeframe in which the data associated with the instances was created.

The conception of the dataset spanned the course of roughly two months. The data generation itself took on the scale of several hours on consumer grade hardware.

**Were any ethical review processes conducted (e.g., by an institutional review board)?** If so, please provide a description of these review processes, including the outcomes, as well as a link or other access point to any supporting documentation.

No, the entire dataset is purely synthetic and only consists of geometric objects, so no ethical review was required.

**Does the dataset relate to people?** If not, you may skip the remaining questions in this section.

No.

**Did you collect the data from the individuals in question directly, or obtain it via third parties or other sources (e.g., websites)?**

No.

**Were the individuals in question notified about the data collection?** If so, please describe (or show with screenshots or other information) how notice was provided, and provide a link or other access point to, or otherwise reproduce, the exact language of the notification itself.

No.

**Did the individuals in question consent to the collection and use of their data?** If so, please describe (or show with screenshots or other information) how consent was requested and provided, and provide a link or other access point to, or otherwise reproduce, the exact language to which the individuals consented.

No.

**If consent was obtained, were the consenting individuals provided with a mechanism to revoke their consent in the future or for certain uses?** If so, please provide a description, as well as a link or other access point to the mechanism (if appropriate).

No.

**Has an analysis of the potential impact of the dataset and its use on data subjects (e.g., a data protection impact analysis) been conducted?** If so, please provide a description of this analysis, including the outcomes, as well as a link or other access point to any supporting documentation.

No.

**Any other comments?**

No further comments.

---

**Preprocessing/cleaning/labeling**

---

**Was any preprocessing/cleaning/labeling of the data done (e.g., discretization or bucketing, tokenization, part-of-speech tagging, SIFT feature extraction, removal of instances, processing of missing values)?** If so, please provide a description. If not, you may skip the remainder of the questions in this section.

No preprocessing was involved in the creation of this dataset.

**Was the "raw" data saved in addition to the preprocessed/cleaned/labeled data (e.g., to support unanticipated future uses)?** If so, please provide a link or other access point to the "raw" data.

Since no preprocessing was conducted, the "raw" data and its label are the primary content of this dataset.

**Is the software used to preprocess/clean/label the instances available?** If so, please provide a link or other access point.

No preprocessing was involved in the creation of this dataset.

**Any other comments?**

No further comments.

| Uses |
| :---: |

**Has the dataset been used for any tasks already?** If so, please provide a description.

Yes, we conducted an evaluation on *continual confounding*. Here, we showed that machine learning models get confounded on the `disjoint` data. On the `strict` data, we found out that performance on the unconfounded data is better when training on data from all confounded tasks jointly when compared to the continual learning setup.

**Is there a repository that links to any or all papers or systems that use the dataset?** If so, please provide a link or other access point.

There is no such repository at the time of the creation of this dataset sheet.

**What (other) tasks could the dataset be used for?**

The primary purpose of ConCon is the investigation of *continual confounding*, but it could also be used for other tasks, such as neuro-symbolic continual learning.

**Is there anything about the composition of the dataset or the way it was collected and preprocessed/cleaned/labeled that might impact future uses?** For example, is there anything that a future user might need to know to avoid uses that could result in unfair treatment of individuals or groups (e.g., stereotyping, quality of service issues) or other undesirable harms (e.g., financial harms, legal risks) If so, please provide a description. Is there anything a future user could do to mitigate these undesirable harms?

None that the authors are aware of, due to the synthetic nature of the dataset consisting of simulated simple objects (like colored cubes and squares) coupled with logical rules.

**Are there tasks for which the dataset should not be used?** If so, please provide a description.

Please refer to the previously asked similar question.

**Any other comments?**

No further comments.

| Distribution |
| :---: |

**Will the dataset be distributed to third parties outside of the entity (e.g., company, institution, organization) on behalf of which the dataset was created?** If so, please provide a description.

Yes, the dataset is publicly available at Zenodo and the source code is available at Github.

**How will the dataset will be distributed (e.g., tarball on website, API, GitHub)** Does the dataset have a digital object identifier (DOI)?

We use the third-party open-access repository Zenodo[9]. Zenodo features unique digital object identifiers. The Zenodo DOI for our dataset is: `10.5281/zenodo.10630482`.

**When will the dataset be distributed?**

The dataset was made available on 9th February, 2024.

**Will the dataset be distributed under a copyright or other intellectual property (IP) license, and/or under applicable terms of use (ToU)?** If so, please describe this license and/or ToU, and provide a link or other access point to, or otherwise reproduce, any relevant licensing terms or ToU, as well as any fees associated with these restrictions.

The dataset is distributed under the BSD 2-Clause "Simplified" License.

**Have any third parties imposed IP-based or other restrictions on the data associated with the instances?** If so, please describe these restrictions, and provide a link or other access point to, or otherwise reproduce, any relevant licensing terms, as well as any fees associated with these restrictions.

No.

**Do any export controls or other regulatory restrictions apply to the dataset or to individual instances?** If so, please describe these restrictions, and provide a link or other access point to, or otherwise reproduce, any supporting documentation.

No.

**Any other comments?**

No further comments.

| **Maintenance** |
|:---:|

**Who will be supporting/hosting/maintaining the dataset?**

We use Zenodo to host the dataset, a reputable open-access repository where the dataset has a high likelihood to persist long into the future. Maintaining is carried out by the authors and affiliated groups.

**How can the owner/curator/manager of the dataset be contacted (e.g., email address)?**

The creators of the datasets: Florian Peter Busch and Roshni Kamath can be contacted at florian_peter.busch@tu-darmstadt.de and roshni.kamath@tu-darmstadt.de respectively.

**Is there an erratum?** If so, please provide a link or other access point.

There presently is no erratum. However, Zenodo offers an erratum functionality should it become relevant in the future.

**Will the dataset be updated (e.g., to correct labeling errors, add new instances, delete instances)?** If so, please describe how often, by whom, and how updates will be communicated to users (e.g., mailing list, GitHub)?

Currently, no immediate updates are envisioned.

---

[9]https://zenodo.org/

**If the dataset relates to people, are there applicable limits on the retention of the data associated with the instances (e.g., were individuals in question told that their data would be retained for a fixed period of time and then deleted)?** If so, please describe these limits and explain how they will be enforced.

The dataset does not relate to people.

**Will older versions of the dataset continue to be supported/hosted/maintained?** If so, please describe how. If not, please describe how its obsolescence will be communicated to users.

Yes, Zenodo supports access of old versions using unique Digital Object Identifiers (DOI).

**If others want to extend/augment/build on/contribute to the dataset, is there a mechanism for them to do so?** If so, please provide a description. Will these contributions be validated/verified? If so, please describe how. If not, why not? Is there a process for communicating/distributing these contributions to other users? If so, please provide a description.

The code for data generation is publicly available on Github at `https://github.com/ml-research/concon`. Validation/Verification of future contributions will not be in the scope of the authors.

**Any other comments?**

No further comments.

