# OpenReview forum: "Where is the Truth? The Risk of Getting Confounded in a Continual World"
_ICML.cc/2025/Conference — ICML 2025 spotlightposter_

### Official Review · Reviewer_iRoy · 2025-03-06

**Overall Recommendation:** 3

**Summary:**

The paper explores a nuanced aspect of continual learning related to confounding data. It highlights how confounding data can create shortcuts by fostering spurious correlations, ultimately hindering the generalization ability of continual learning methods. The authors demonstrate the effects of confounding data on sequential continual learning using a confounding dataset generated from CLEVR. Their findings show that conventional continual learning methods struggle in this setting, underscoring the need for more robust approaches.

**Claims And Evidence:**

The authors claim to study continual confounding using ConCon dataset and they also provide evidence supporting the claim but evidence does seem strong since it lacks evidence on other real world datasets.

**Essential References Not Discussed:**

References are adequate.

**Experimental Designs Or Analyses:**

Yes

**Methods And Evaluation Criteria:**

Yes

**Other Comments Or Suggestions:**

None Found

**Other Strengths And Weaknesses:**

## Strengths

- The paper is well-written and easy to follow.
- It offers a novel perspective on continual learning by focusing on the impact of confounding data, rather than the commonly studied issue of catastrophic forgetting. This adds value to the field, where most literature centers around catastrophic forgetting.

## Weaknesses
- The ConCon dataset, used to study the impact of confounding data, appears somewhat artificial and monotonous.
- While the paper effectively identifies a critical challenge in continual learning, it does not propose any solutions to address it.

**Questions For Authors:**

While the paper presents a novel perspective on continual learning, it lacks strong evidence of continual confounding due to the absence of evaluations on real-world datasets. Additionally, it does not propose any solutions, leaving the community without a clear direction for addressing and mitigating the identified issue.

- Is it possible to evaluate on the dataset similar to the figure 1 shown in the paper?
- Why the authors have not investigated the solution of the problem described in the paper?

**Relation To Broader Scientific Literature:**

Majority of the Continual Learning literature focuses on issue of Catastrophic Forgetting while this paper focuses on continual confounding which adds value to this field.

**Theoretical Claims:**

Yes

---

> ### Author Rebuttal · Authors · 2025-04-01
>
> We thank the reviewer for their feedback. In the following, we would like to address their two concerns.
>
> **Synthetic vs Real-World Datasets**
>
> In our paper, we introduce a real-world experiment on the popular ImageNet dataset and observe a surprising discrepancy between joint and cumulative training. By introducing the ConCon dataset, we are able to systematically investigate variants of continual confounding and eliminate other (potentially unknown) factors present in real-world images. This enables us to eliminate alternative causes of unexpected model behavior and allows for analyzing the effects of continual confounding in isolation.
>
> Our real-world experiment introduced in section 1 shows the same behavior as we observed in our experiments on ConCon: cumulative training results in a lower accuracy than joint training. In order to further highlight that this behavior can also occur on natural images, we also include another real-world experiment involving a higher number of ImageNet classes. Here, we even observe a 5% difference between cumulative and joint training. Please consider the experimental details and results below ("New Real-World Experiment on ImageNet"). We have included this experiment in the appendix.
>
> **New Real-World Experiment on Imagenet**
>
> In this experiment, we select several ImageNet classes such that in each of the three tasks, we have a group of 4 ImageNet classes representing 4-legged animals and 4 ImageNet classes not representing 4-legged animals. Each group of 4 classes is assigned one label, resulting in 6 labels for the model to learn. See the table below for the class selection. The resulting groups are correlated with colors (Task 1: white vs blue, Task 2: green vs white, Task 3: blue vs non-blue). We hypothesize that for each task in isolation, learning to make predictions based on the confounding feature "color" is easier than learning the ground-truth features that correspond to "animal with 4 legs".
>
> |                     Task 1                      |                    Task 2                     | Task 3                                  |
> | --------------------------------------------- | ------------------------------------------- | --------------------------------------- |
> | arctic fox, polar bear, white wolf, samoyed dog |         cheetah, deer, lion, leopard          | hippo, crocodile, water buffalo, beaver |
> |     jellyfish, blue jay, blue shark, tench      | snowmobile, sailboat, snowplough, rattlesnake | eagle, spider, lifeboat, mushroom       |
>
> As for our other experiments, we ran cumulative and joint training on a Resnet-18 model and averaged the results of 5 different seeds. The results for joint training and for cumulative training after covering all three tasks are shown in the table below:
>
> | Cumulative | 76.3%  | 78.5%  | 84.4%  |
> | ---------- | ------ | ------ | ------ |
> | Joint      | 83.45% | 86.25% | 86.25% |
>
> Across all tasks, we observe an average decrease in accuracy of 5.58 percentage points, a substantial degradation in performance. This provides further evidence that continual confounding can manifest in real-world images and that it does deteriorate model performance. This experiment has been added to the appendix.
>
> **Contribution Despite a Lack of Solutions for Continual Confounding**
>
> This paper is the first to investigate continual confounding. We not only showcase that this is an issue on real-world data but also introduce a thorough framework that provides a foundation for systematically studying different variants of confounding based on CLEVR, where one has complete control over the distribution of the generated images. ConCon, the resulting benchmark on continual confounding, is, therefore, one of many dataset and benchmark papers that are common in the machine learning community and fall under the ICML call for papers as "datasets that are of interest to the machine learning community". Our findings include a discussion on insidious continual confounding, a phenomenon where cumulative training yields worse accuracies than joint training. This by itself is an important contribution, as it questions the use of cumulative training as the gold-standard upper-bound for evaluating continual learning methods, which is a common approach in continual learning research.
>
> We fully agree that developing methodological solutions for overcoming our identified issue of continual confounding is very important. However, we consider this to be out of the scope of this dataset paper, which aims to identify and analyze the issue. We believe that ConCon will be of great use in evaluating such novel approaches. However, doing so requires further research and more in-depth analysis than could be provided by this paper.

---

### Official Review · Reviewer_KoqF · 2025-03-13

**Overall Recommendation:** 3

**Summary:**

This paper introduces the concept of continual confounders. A data contains confounders when a model trained on the data can fit the training data using spurious correlations but fails to generalize at test time. Continual confounders are ones that control distributions across a continual set of tasks. The paper also introduces a dataset, ConCon, a simulated dataset built on the framework of CLEVR using Blender software consisting of images of various objects with varying textures and colors in a 3D space.

Section 3 introduces the concept of continual confounding. Continual confounders are divided into two categories, disjoin and strict. Disjoint confounders are only observed in their respective tasks. A model which fits continually disjoint confounders need not to unlearn the confounders of previous tasks to learn a new task. Strict confounders on the other hand, may appear in other tasks. Section 3.1 provides a rigorous definition of confounders. Section 3.2 introduces the ConCon dataset that contains disjoint and strict variants. The dataset contains various objects such as spheres, cubes, and cylinders in different sizes, colors, and textures. The task is binary classification to determine whether a sphere and small cube exists in the image. The confounders are the blue, metal, and large.

It is argued that static confounders are insidious continual confounders that if a model is trained jointly on the dataset, it can generalize well by not utilizing confounders but learned continually, it will fail to generalize. As such avoiding catastrophic forgetting is not enough for continual learners in such a setup.

The experiments evaluate the performance of continual learning methods for training two architectures, ResNet-18, and NeSy. The continual learning methods include replay methods and regularization methods. The following observations are made:
- All CL methods on both models fail on unconfounded held-out test sets on the disjoint variant of ConCon. Similarly on the static variant all methods except for joint and cumulative training do not generalize.
- Preventing catastrophic forgetting on the disjoint dataset does not help with making correct predictions on unconfounded data.
- Training in a continual setup performs significantly worse than the joint setup.
- Continual learning methods suffer from insidious continual confounding.

## Update after rebuttal
I thank authors for their response including clarifications and new results with PNNs. I recommend incorporating the response into a revision. I maintain a weak accept rating for this work as I believe this dataset may be valuable for the community in studying continual learning methods but understand other reviewers have remaining concerns about the dataset being synthetic.

**Claims And Evidence:**

The paper makes the following claims and provide evidence in a synthetic setup:
- Continual learning methods suffer from insidious continual confounding where the model can generalize if trained jointly on the data but does not generalize when trained continually.
- Preventing catastrophic forgetting does not help with making correct predictions on unconfounded data.
- The ConCon dataset is a benchmark for evaluating the impact of confounders in continual learning.

**Essential References Not Discussed:**

The paper is missing references to works on spurious correlations and related benchmarks. For example:
- Koh, P. W., et al., Wilds: A benchmark of in-the-wild distribution shifts. ICML 2021.
- Sagawa, S., et al. Distributionally robust neural networks for group shifts: On the importance of regularization for worst-case generalization. ICLR 2020.

A discussion and comparison is needed, especially in terms of the types of spurious correlations identified in prior works.

**Experimental Designs Or Analyses:**

The construction of the ConCon dataset with disjoint and strict variants is sound. The results on these datasets also match the intuition.

**Methods And Evaluation Criteria:**

The paper does not propose a new method. It provides a new evaluation benchmark that controls the confounders over tasks and provides meaningful observations.

**Other Comments Or Suggestions:**

Typos:
- Line 41: Resnet -> ResNet
- Line 86: a -> an
- Line 120: introduction example -> introductory example
- Section 3: it may be helpful to add a summary table for comparison between disjoint and strict continual confounders and some of their characteristics. For example, which learner (with or without replay) in which setup (continual vs jointly) on which confounders (strict, disjoint) can fit/generalize.
- Line 254: comprising of -> comprises

**Other Strengths And Weaknesses:**

Strengths:
- The paper introduces an interesting concept of continually confounded datasets, rigorously defines them and introduces an evaluation that provides novel insights.

Weaknesses:
- It is not clear whether the goal of the benchmark is to inspire model architecture design in the future or better continual training methods or to suggest some continual tasks are not learnable by continual learners. The paper needs to clarify its goal.
- The evaluation focuses on only two models, ResNet-18 and NeSy trained from scratch on the data. For example, what happens if a pretrained model is fine-tuned on the data? What are the implications for continual learning methods with adaptive architectures such as Progressive Neural Networks (Rusu, Andrei A., et al. "Progressive neural networks." (2016))?

**Questions For Authors:**

- Is insidious continual confounding limited to strict confounders? Or is insidious continual confounding a property that various confounder types can have including strict confounders?

**Relation To Broader Scientific Literature:**

This work is related to two literatures on continual learning and spurious correlations. The connection between these two literatures has not been explored before and is interesting to study.

**Theoretical Claims:**

The paper provides a rigorous definition of confounders but no theoretical claims.

---

> ### Author Rebuttal · Authors · 2025-04-01
>
> We thank the reviewer for their feedback and positive evaluation of our work. We would like to comment on a few points in their review.
>
> **Related Work on Distribution Shifts**
>
> Thank you for suggesting further related work. We have included the references and discussed their relation to ConCon and the goal of our paper.
>
> **Goal of This Paper and ConCon**
>
> The goal of our paper is to "improve the evaluation of continual learning methods when confounding might be present and also inspire new approaches to deal with these issues. The ConCon benchmark is designed to facilitate systematic investigation of such methods in the future." We included these sentences in our paper. The disjoint dataset represents a simple confounded setting where confounding features only appear on certain images. A good example of such confounding is watermarks. We show that a challenge for CL methods on disjoint datasets is not only to avoid forgetting but also to learn the ground-truth features and not just the disjunction of confounders. The confounding in our strict dataset is less obvious, as confounding features may appear as regular, random features in other tasks. For example, snowy backgrounds might at first only be associated with polar bears, but in later classes, they also appear on other images in winter. Perhaps surprisingly so, we show that this type of insidious continual confounding can deteriorate model performance compared to non-continual learning settings. Our goal is, thus, to highlight these challenges and dangers related to confounding in continual settings. Our results on insidious continual confounding also question the use of cumulative training as the gold-standard upper-bound in continual learning.
>
> Our response to reviewer iRoy about our contributions might also be of interest.
>
> **Effects of Pre-Training and Implications on Adaptive Architectures**
>
> Our NeSy model is pre-trained on the CLEVR dataset. However, we see that it does not perform well on the unconfounded datasets, where the accuracies are worse than for the Resnet-18 model (70.8 vs 95.7). Pretraining, in general, can both be helpful and harmful when it comes to continual confounding, depending on what the model is trained on. If the pre-trained model has learned the ground-truth prediction, it is unlikely to unlearn that even on confounded data. However, our comparison of joint and cumulative training shows how a model pre-trained (tasks 1 and 2) to focus on confounders performs worse than a model trained from scratch (joint training).
>
> Following the reviewer's suggestion, we ran the Progressive Neural Networks (PNNs) method on the two variants of our ConCon dataset for the NN model and added them to tables 1 and 2 :
>
> |          | Task 1 | Task 2 | Task 3 | Task 1 @ Task 3 | Task 2 @ Task 3 | UnConf |
> | -------- | ------ | ------ | ------ | --------------- | --------------- | ------ |
> | Strict   | 100.0  | 99.77  | 99.76  | 100.0           | 99.7            | 49.2   |
> | Disjoint | 100.0  | 99.84  | 100    | 100.0           | 99.6            | 47.89  |
>
> We expand the ResNet-18 model when it encounters new tasks.
> Data from confounded tasks always use their respective output heads, and the unconfounded data uses the most recent head (task 3).
> Generally speaking, we do not expect adaptive architectures to help against continual confounding by themselves. Since the model first learns to focus on confounding features, thereby learning simple solutions and setting model weights accordingly, expanding the architecture does not automatically help learn the correct, more complex ground-truth rule. We hypothesize that such a model must be able to unlearn the previously learned behavior in order to ignore the confounders and learn the ground-truth feature instead. Therefore, we argue that adaptive architectures are unlikely to help against continual confounding, although specialized approaches might be useful.
>
> **Question on Insidious Continual Confounding**
>
> > Is insidious continual confounding limited to strict confounders? Or is insidious continual confounding a property that various confounder types can have including strict confounders?
>
> Confounders / confounding features themselves do not exhibit insidious continual confounding. Insidious continual confounding arises when a model is continually trained on confounded data, where confounding causes it to make predictions with lower accuracy than in a joint training scenario. (Note that confounding that does not change across tasks would not cause insidious continual confounding as joint training would suffer from it in the same way; see our evaluation on the disjoint dataset.) Whether insidious continual confounding occurs thus depends on both the type of confounding and the model and training procedure.
>
> **Typos**
>
> We thank the reviewer for their feedback on typos. We will correct them for the camera-ready version.

---

> > ### Comment · Reviewer_KoqF · 2025-04-03
> >
> > I thank authors for their response including clarifications and new results with PNNs. I recommend incorporating the response into a revision. I maintain a weak accept rating for this work as I believe this dataset may be valuable for the community in studying continual learning methods but understand other reviewers have remaining concerns about the dataset being synthetic.

---

### Official Review · Reviewer_yrkD · 2025-03-14

**Overall Recommendation:** 4

**Summary:**

The paper presents the confounder problem in the continual learning regime, which is novel. It also establishes a benchmark with clear logical definitions and potentially highlights a new direction for studies in the continual learning field to improve the overall performance.

**Claims And Evidence:**

The claims made in the work are clear and logical.

**Essential References Not Discussed:**

Here are some related continual learning works the authors may consider including:

[1] Liu, Y., Zhu, W., & Ren, S. (2022). Navigating memory construction by global pseudo-task simulation for continual learning. Advances in Neural Information Processing Systems, 35, 7342-7355.

[2] Farajtabar, M., Azizan, N., Mott, A., & Li, A. (2020, June). Orthogonal gradient descent for continual learning. In International conference on artificial intelligence and statistics (pp. 3762-3773). PMLR.

**Experimental Designs Or Analyses:**

The experimental design is scientific, and the related continual learning methods are evaluated on the new benchmarks.

**Methods And Evaluation Criteria:**

The evaluation methods presented in this work are novel and valid for use in continual learning scenarios to investigate whether the model is truly learning in a sequential setting rather than relying on shortcuts.

**Other Comments Or Suggestions:**

More models could be tested on the dataset for evaluation, but it is not necessary. The ResNet and NeSy models based on transformers used in the experiments are sufficient to support the statements.

**Other Strengths And Weaknesses:**

The work is well-written and complete. The continual confounding is clearly defined through logical rules and effectively explained with text and images. The dataset benchmark, ConCon, constructed also makes sense.

**Questions For Authors:**

None so far.

**Relation To Broader Scientific Literature:**

N/A

**Theoretical Claims:**

Not applicable.

---

> ### Author Rebuttal · Authors · 2025-04-01
>
> We thank the reviewer for their positive feedback and we will include the suggested references in our related work section. We are happy to read that they agree that our experiments are sufficient to support our claims. Nevertheless, we invite the reviewer to also take a look at our responses to the other reviewers, where we add additional experiments that support our claims even further.

---

### Official Review · Reviewer_SVA5 · 2025-03-15

**Overall Recommendation:** 3

**Summary:**

This paper explores confounding in continual learning. The authors formally describe confounding factors that lead to poor generalization and introduce a CLEVR-based synthetic dataset (ConCon) to study these challenges. They evaluate several continual learning approaches on ConCon and show that these methods struggle to mitigate the influence of confounders, leading to degraded performance on unconfounded test data.

### Update after rebuttal
The author's rebuttal addresses most of my concerns. Thus I have updated my score from 2 to 3.

**Claims And Evidence:**

1. Sequential vs. Joint Training:
The paper claims that sequential training (including cumulative training) is more challenging in the presence of confounders compared to joint training. Figure 1 and Table 2 indicate that joint training outperforms cumulative training. However, this may be due to the inherent difficulty of training on streaming data—non-convex optimization can lead to overfitting on biased, limited data, causing poor local minima. Thus, the performance gap might primarily reflect the challenge of learning from changing distributions rather than the effect of confounders alone.

2. Generalization in Continual Learning:
The authors argue that even methods that prevent forgetting in continual learning fail to generalize well due to confounders, as shown in Table 1. While this observation may be valid, the experimental design appears to mix the issue with a train-test distribution mismatch. In task t, the training data includes both ground-truth features (g) and task-specific confounders (c_t), but the test set only contains the ground-truth feature g. This mismatch might be driving the performance gap, which is more an issue of domain adaptation than a failure of continual learning.

**Essential References Not Discussed:**

Including related work from the domain generalization literature—where similar challenges of spurious correlations are addressed—would help strengthen the context for the paper’s contributions.

**Experimental Designs Or Analyses:**

The experimental evaluation is based on the ConCon dataset. Concerns include:
1. Task Design: The current task design may not adequately isolate the effect of confounders from the inherent difficulty of learning on non-stationary data.
2. Train-Test Mismatch: The performance gap might be largely due to the mismatch between training (with both g and c_t) and testing (with only g), which is more a domain generalization issue than a continual learning problem.
3. Limited Real-World Evaluation: Relying exclusively on a synthetic dataset raises questions about the generalizability of the findings to real-world scenarios.

**Methods And Evaluation Criteria:**

The paper primarily introduces a new concept and dataset rather than novel methods. It proposes a formal description of confounding and uses the ConCon dataset—with its controllable confounding factors—to evaluate various continual learning approaches. However, the evaluation might be skewed by the train-test mismatch, making it unclear if the poor generalization is due to continual learning shortcomings or simply the data distribution shift.

**Other Comments Or Suggestions:**

1. The memory size used in experiments (e.g., 100 samples) may be too small to reflect practical scenarios; larger memory sizes should be considered, especially in table 2.

2. If the goal is to evaluate model performance under different distributions, exploring domain adaptation techniques might be more appropriate than a pure continual learning approach.

**Other Strengths And Weaknesses:**

Strengths:
1. Studies an interesting and underexplored problem in continual learning, which is related to dataset confounding.
2. Introduces a novel, controlled synthetic dataset (ConCon) to study confounding effects.

Weaknesses:
1. The paper needs a clearer distinction between ground-truth features and confounders. If a factor is deemed confounding, it should ideally be identifiable in both the training and test sets; otherwise, the evaluation might be misleading.
2. The design suffers from a train-test mismatch that may be driving the observed performance gaps. The performance differences in Figure 1 could stem from challenges associated with streaming data rather than solely from the presence of confounders. Consider redesigning the experimental tasks (e.g., grouping tasks by different background colors) to better isolate the impact of confounders. Specifically, Task 1 could involve classification between images with a blue background (e.g., jellyfish and shark), while Tasks 2 and 3 could focus on white and green backgrounds, respectively. Comparing the final performance between this revised design and the original design would help assess the influence of confounding factors.
3. The reliance on the proposed synthetic dataset limits the applicability of the findings. Evaluating on real-world or multi-class datasets would provide stronger evidence for the claims.
4.  It is helpful to incorporate more baselines. Such as generative reply [1], which is also an important baseline in continual learning.

[1] Van de Ven, Gido M., Hava T. Siegelmann, and Andreas S. Tolias. "Brain-inspired replay for continual learning with artificial neural networks." Nature communications 11.1 (2020): 4069.

**Questions For Authors:**

1. How do you ensure a clear and consistent distinction between ground-truth features and confounders in the ConCon dataset?
2. Could the performance gap between joint and sequential training be primarily attributed to the challenges of streaming data (non-convex optimization and local minima) rather than confounders?
3. How do you envision the proposed framework and findings translating to real-world continual learning scenarios, where data distributions might differ less drastically between training and testing?

For additional details on potential weaknesses and suggested modifications, please refer to the previous discussion.

**Relation To Broader Scientific Literature:**

The paper situates its contributions within the continual learning literature. It would benefit from a deeper discussion on the relationship between domain generalization and invariant risk minimization literature, both of which address issues of spurious correlations and distribution shifts.

**Theoretical Claims:**

The paper presents a formal description of continual confounding using Boolean algebra to capture the interplay between ground-truth predicates and task-specific confounders. Although the theoretical framework is plausible, the assumptions require further clarification. In particular, the distinction between ground-truth features and confounders is not well defined; defining confounders solely based on the test set is problematic.

---

> ### Author Rebuttal · Authors · 2025-04-01
>
> We thank the reviewer for the detailed and constructive feedback. We first try to clarify their more general concerns.
>
> Features that act as confounders in one dataset can also appear as random features in other datasets. The distribution shift between the confounded tasks and the unconfounded dataset is a result of such features no longer **acting** as confounders, i.e., they might appear on both positive and negative images but are not informative wrt the class label. Our synthetic data generation used for ConCon ensures minimal distribution shift between confounded and unconfounded datasets while still exhibiting confounding in one but not the other. Thus, the resulting minimal train-test mismatch is intentional and allows for investigating whether a high model accuracy is caused by the model making predictions based on the confounding features or the ground-truth features. We have included this explanation in section 3.2.
>
> In addition to experience replay (ER), where we use a buffer of 100 samples, we also evaluate ER with an infinite buffer size. This is what we refer to as *cumulative training*. Any other buffer size larger than 100 should, therefore, result in accuracies between those of ER and cumulative training. We made changes to clarify this in Section 4.1.
>
> We will now respond to the weaknesses (W) and questions (Q).
>
> **W1/Q1**
>
> In this paper, there is no a priori distinction between confounding features and ground truth features. This is intentional, as we aim to investigate the models' capabilities in identifying the ground truth features without prior information. As a result, confounding is defined relative to the full set of tasks. How features that act as confounders in one task appear in other tasks depends on the dataset variant:
> In the disjoint dataset, confounding features are only present in their respective task. They are identifiable in the sense that their absence in the other tasks makes them irrelevant and unhelpful for accurate prediction-making. In the strict dataset, confounding features appear in other tasks in both positive and negative images. They are, therefore, incompatible with the correct decision rule in new tasks.
>
> Any confounding features of the confounded tasks may appear in images of either class in the unconfounded dataset. However, they are uninformative and provide no value for prediction-making, as they do not act as confounders there.
>
> **W2**
>
> We ran experiments on ConCon where we shuffled the training data before distributing it across tasks, thereby removing the alignment of confounders:
>
> |          | Task 1 | Task 2 | Task 3 | Task 1 @ Task 3 | Task 2 @ Task 3 | UnConf |
> | -------- | ------ | ------ | ------ | --------------- | --------------- | ------ |
> | Strict   | 79.85  | 93.4   | 95.32  | 95.18           | 96.72           | 89.52  |
> | Disjoint | 99.98  | 99.90  | 99.96  | 99.98           | 99.84           | 49.97  |
>
> For strict, the accuracy on UnConf is much closer to joint (95.7) than to cumulative (72.6). This confirms that insidious continual confounding is responsible for a majority of the drop in accuracy.
>
> For the ImageNet experiment, we align the confounders task-wise: Task 1: Arctic fox, snowmobile, Task 2: broccoli, tree frog, Task 3: tiger shark, jellyfish.
>
> We obtained 89.93% joint and 89.0% cumulative accuracy. Unfortunately, any differences here are obscured by variance due to data loading order, so we cannot draw substantial conclusions.
>
> **Q2**
>
> The gap arises because of the combination of confounders and the challenges of streaming data. As we also show in our new experiment on ConCon above, the gap between cumulative and joint training is more substantial when tasks are confounded.
>
> **W3**
>
> We ran additional experiments on real-world data. Please refer to our response to reviewer iRoy.
>
> **Q3**
>
> In real-world scenarios, we expect confounding to be imperfect. This could express itself in two ways:
>
> 1. Only a subset of the data is confounded.
> 2. The entire dataset is imperfectly confounded, i.e., in the training set, the confounders provide some limited bits of information about the class, but this relationship does not generalize on the test set.
>
> **W4**
>
> We here show the results for generative replay:
>
> |          | Task 1 | Task 2 | Task 3 | Task 1 @ Task 3 | Task 2 @ Task 3 | UnConf |
> | -------- | ------ | ------ | ------ | --------------- | --------------- | ------ |
> | Strict   | 99.96  | 98.54  | 99.58  | 48.84           | 48.61           | 49.26  |
> | Disjoint | 100.0  | 88.78  | 99.28  | 53.2            | 59.33           | 47.73  |
>
> Results match those reported for other approaches in our paper. We have included it in the appendix.
> Please let us know if the reviewer recommends any other methods to include.

---

### Decision · Program_Chairs · 2025-05-01

**Decision:**

Accept (spotlight poster)

**Comment:**

This paper received three weak accepts and one accept, reflecting generally positive reception with some reservations. The work addresses an important and underexplored issue in continual learning, dataset confounding, and introduces a novel synthetic benchmark to systematically study these effects. The paper is well-written and the continual confounding scenarios are clearly defined and illustrated. However, reviewers noted areas for improvement, including the need for a clearer separation between ground-truth features and confounders, potential train-test mismatches affecting results, and the limitations of relying solely on synthetic data. Suggestions were made to include additional baselines and to evaluate real-world datasets to improve generalizability. Overall, the dataset and framework have the potential to be a valuable tool for the community, and the authors are encouraged to incorporate reviewer feedback in the final version.